Assessing the roles emission sources and atmospheric processes play in simulating
$\delta^{15}N$ of atmospheric $NO_x$ and $NO_3^-$ using CMAQ (version 5.2.1) and SMOKE
(version 4.6).
*Huan Fang[†] and Greg Michalski[†‡]*
[†]Department of Earth, Atmospheric, and Planetary Sciences Purdue University, 550 Stadium Mall
Drive, West Lafayette, IN 47907, United States
[‡]Department of Chemistry, Purdue University, 560 Oval Drive, West Lafayette, IN 47907, United
States
Correspondence: Huan Fang, fang63@purdue.edu
Keywords: isotope, nitrogen, atmospheric NOx, atmospheric nitrate, NOx emission sources,
emission inventory, emission input dataset, atmospheric processes, disperse, mixing, transport,
chemical transport model, 3D CTM, NEI, SMOKE, CMAQ
Abstract
Nitrogen oxides ($NO_x$ = nitric oxide (NO) + nitrogen dioxide ($NO_2$)) are important trace gases that
affect atmospheric chemistry, air quality, and climate. Contemporary development of $NO_x$
emissions inventories is limited by the understanding of the roles of vegetation (net $NO_x$ source or
net sink), gasoline and diesel in vehicle emissions, and the application of $NO_x$ emission control
technologies. The nitrogen stable isotope composition ($\delta^{15}N$) of $NO_x$ is an effective tool to evaluate
the accuracy of the $NO_x$ emission inventories, which are based on different assumptions. In this
study, we traced the changes in $\delta^{15}N$ values of $NO_x$ along the "journey" of atmospheric $NO_x$, driven
by atmospheric processes after different sources emit $NO_x$ to the atmosphere. The $^{15}N$ was
incorporated into the emission input dataset, generated from the US EPA trace gas emission model
SMOKE (Sparse Matrix Operator Kernel Emissions). Then the $^{15}N$ incorporated emission input
dataset was used to run CMAQ (the Community Multiscale Air Quality Modeling System). The
simulated spatiotemporal patterns in $NO_x$ isotopic composition for both SMOKE outputs and
CMAQ outputs were compared with corresponding atmospheric measurements in West Lafayette,
Indiana, USA. By enhancing $NO_x$ deposition, we simulated the expected $\delta^{15}N$ of $NO_3^-$ assuming
no isotope fractionation during chemical conversion or deposition. These simulations were
compared to $\delta^{15}N$ of $NO_3^-$ in NADP sites. The results indicate the potential underestimation of
emissions from soil, livestock waste, off-road vehicles, and natural gas power plants and the
potential overestimation of emissions from on-road vehicles and coal-fired power plants, if only
considering the difference in $NO_x$ isotopic composition for different emission sources. The
estimation of atmospheric $\delta^{15}N(NO_x)$ using CMAQ shows better agreement (by ~3‰) with
observations than using SMOKE (Sparse Matrix Operator Kernel Emissions), due to the
consideration of mixing, dispersion, transport, and deposition of $NO_x$ emission from different
sources.

1.  Introduction

NO$_x$ are important trace gases that affect atmospheric chemistry, air quality, and climate (NO$_x$ = NO + NO$_2$). The main sources of tropospheric NO$_x$ are emissions from vehicles, power plants, agriculture, livestock waste, as well as the natural by-product of nitrification and denitrification occurring in soil, and lightning (Galloway, et al., 2004). The NO$_x$ photochemical cycle generates OH and HO$_2$ radicals, organic peroxy radicals (RO$_2$), and ozone (O$_3$), which ultimately oxidize NO$_x$ into NO$_y$ (NO$_y$ = NO$_x$ + HONO + HNO$_3$ + HNO$_4$ + N$_2$O$_5$ + other N oxides). During the photochemical processes that convert NO$_x$ to NO$_y$, ground-level concentrations of O$_3$ become elevated and secondary particles are generated. Secondary aerosols in turn affect cloud physics, enhancing the reflection of solar radiation (Schwartz, 1996) and are hazardous to human health (Lighty et al., 2000). Thus, the importance of NO$_x$ in air quality, climate, and human and environmental health makes understanding the spatial and temporal variation in the sources of NO$_x$ a vital scientific question.

Despite years of research, however, there are still several significant uncertainties in the NO$_x$ budget. About 15% of global NO$_x$ emissions, ranging from 6.6 to 21 Tg N yr$^{-1}$, is derived from global soil NO$_x$ emissions yet evaluating and verifying emission rates using both laboratory and field measurements is still a challenge (Jaeglé et al., 2005; Yan et al., 2005; Stehfest and Bouwman, 2006; Hudman et al., 2012). Soil NO$_x$ emissions vary by different biome types, meteorological conditions, and soil physicochemical properties. The application of N fertilizer also has a strong effect on soil NO$_x$ emissions, which can dramatically increase during the first 1-2 days after N fertilizer application and can take several weeks for the emission rate to drop to pre-fertilizer levels (Ludwig et al., 2001). Furthermore, the role of vegetation, acting as a net source of atmospheric NO$_x$ when ambient NO$_x$ concentration is below the "compensation point", while acting as a net sink of atmospheric NO$_x$ when ambient NO$_x$ concentrations are above it (Johansson, 1987; Thoene, Rennenberg & Weber, 1996; Slovik et al., 1996; Webber & Rennenberg, 1996). This significantly impacts the biotic NO$_x$ emission inventory (Almaraz et al., 2018). Uncertainties also exist in the amount of NO$_x$ emitted during the combustion of fossil fuels by vehicles and industry. According to Parrish (2006), the estimation of on-road vehicle NO$_x$ emission has at least 10 to 15% uncertainty. For the mileage-based algorithm, which is used in the National Emission Inventory (NEI), the uncertainty is caused by the limited number of sites to determine the emission factors of vehicle classifications and emission types (Ingalls, 1989; Pierson et al., 1990; Fujita et al., 1992; Pierson et al., 1996; Singer and Harley, 1996). The uncertainty of the alternative fuel-based approach is caused by the fuel sales data and emission factors (Sawyer et al., 2000). The uncertainty in power plant NO$_x$ emissions results from the choice of emission control technologies, of which the removal efficiencies of NO$_x$ emission are different. NO$_x$ removal by low NO$_x$ burning, over-fire air reduction, and selective non-catalytic reduction is highly variable, ranging from 50 to 75% (Srivastava et al., 2005).

The nitrogen stable isotope composition of NO$_x$ might be a useful tool to help resolve the uncertainties of how NO$_x$ emission sources vary in space and time because natural and anthropogenic NO$_x$ sources have distinctive $^{15}$N/$^{14}$N ratios (Ammann et al., 1999; Felix et al., 2012; Felix and Elliott, 2013; Fibiger et al., 2014; Heaton, 1987; Hoering, 1957; Miller et al., 2017; Walters et al., 2015a, 2015b, 2018). This variability in NO$_x$ $^{15}$N/$^{14}$N ratios is quantified by

$$\delta^{15}N(NO_x) \ (‰) = [(^{15}NO_x/^{14}NO_x) / (^{15}N_2/^{14}N_2)_{air} -1] \times 1000) \qquad \text{Eq. (1)}$$

where $^{15}NO_x/^{14}NO_x$ is the measurement of relative abundance of $^{15}N$ to $^{14}N$ in atmospheric $NO_x$,
compared with the ratios in air $N_2 = 0.0036$. Previous research has shown that there are distinctive
differences in $\delta^{15}N$ values for $NO_x$ from different emission sources and significant variations
within each source (Fig. 1). Soil $NO_x$ has the lowest $\delta^{15}N$ values (Li & Wang, 2008; Felix & Elliott,
2014; Yu & Elliott, 2017; Miller et al., 2018) followed by waste (Felix & Elliott, 2014) and $NO_x$
emissions from vehicles (Moore, 1977; Heaton, 1990; Ammann et al., 1999; Pearson et al., 2000;
Savard et al., 2009; Redling et al., 2013; Fibiger, 2014; Felix & Elliott, 2014; Walters et al., 2015a;
Walters et al., 2015b). The $NO_x$ emissions from natural gas power plants are isotopically heavier
than soil and waste (Walters et al., 2015b) while those from coal-fired power plants have the
highest values (Heaton, 1987; Heaton, 1990; Snape, 2003; Felix et al., 2012; Felix et al., 2015;
Savard et al., 2017). The implement of emission control technology tends to increase $NO_x$ $\delta^{15}N$
values in both coal-fired power plants (Felix et al., 2012) and vehicles (Walters et al, 2015a). These
distinctive differences in $\delta^{15}N$ values among different $NO_x$ emission sources suggest $\delta^{15}N$ could
be an effective tracer of atmospheric $NO_x$ sources. For example, Redling et al. (2003) found higher
$\delta^{15}N$ of $NO_2$ in samples collected closer to the highway compared to those adjacent to a forest,
showing the emissions from vehicles were dominant near the highway. A strong positive
correlation between the amount of $NO_x$ emission from coal-fired power plants within 400 km
radial area of study sites and $\delta^{15}N(NO_3^-)$ of deposition has been demonstrated (Elliott et al., 2007;
2009). What is lacking is a systematic way of connecting $\delta^{15}N$ values of $NO_x$ sources, regional
emissions, and data from numerous studies that measure $\delta^{15}NO_y$ (Elliott et al., 2009; Garten, 1992;
Hall et al., 2016; Occhipinti, 2008; Russell et al., 1998).

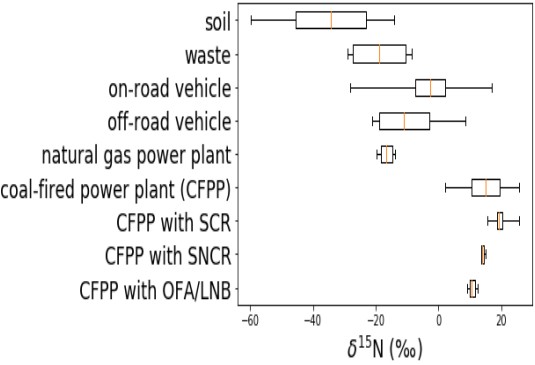

Figure 1: Box (lower quartile, median, upper quartile) and whisker (lower extreme, upper extreme) plot of the distribution of $\delta^{15}N$ values for various $NO_x$ emission sources.

Here we have simulated the emission of $^{15}NO_x$ and its mixing in the atmosphere and compared
the predicted $\delta^{15}N(NO_x, NO_3^-)$ values to observations. The $\delta^{15}N$ values of atmospheric $NO_x$ are
impacted by three main factors. The first is the inherent variability of the $\delta^{15}NO_x$ emissions in time
and space. Secondly, atmospheric processes that mix the emitted $NO_x$, blurring multiple emission
sources within a mixing lifetime relative to the $NO_x$ chemical lifetime (2-7 hours), which depends
on its concentration and photooxidation chemistry, that also vary in time and by location (Laughner
& Cohen, 2019). And thirdly, isotope effects occurring during tropospheric photochemistry may
alter the $\delta^{15}NO_x$ emissions as they are transformed from $NO_x$ into $NO_y$. In this paper, we consider
the effects from the first and second considerations, the temporal and spatial variation in $NO_x$

emission and the impacts from atmospheric transport and deposition processes (source and mixing hypothesis). We accomplish this by incorporating an input dataset of $^{15}$N emissions used in simulations by the Chemistry-Transport Model (CTM) used in CMAQ (The Community Multiscale Air Quality Modeling System). In a companion paper, we will discuss the impacts of tropospheric photochemistry by incorporating a $^{15}$N chemical mechanism (Fang et al., 2021) into CMAQ. The ultimate goal will be to evaluate the accuracy of the NO$_x$ emission inventory using $^{15}$N.

## 2.  Methodology

2.1 Incorporating $^{15}$N into NO$_x$ emission datasets

The EPA trace gas emission model SMOKE (Sparse Matrix Operator Kernel Emissions) was used to simulate $^{14}$NO$_x$ and $^{15}$NO$_x$ emissions. $^{14}$NO$_x$ emissions were estimated using the SMOKE model based on the 2002 NEI (National Emission Inventory, USEPA, 2014), and $^{15}$N emissions were determined using these $^{14}$NO$_x$ emissions and the corresponding $\delta^{15}$N values of NO$_x$ sources from previous research (Table 1). Using the definition of $\delta^{15}$N (‰), $^{15}$NO$_x$ emitted by each SMOKE processing category (area, biogenic, mobile, and point) was calculated by

$$^{15}NO_x(i) = {}^{14}NO_x(i) \times {}^{15}R_{NO_x}(i) \qquad \text{Eq. (2)}$$

where $^{14}$NO$_x$ ($i$) are the NO$_x$ emissions for each category ($i$) obtained from NEI and SMOKE and $^{15}$R$_{NOxi}$ is a $^{15}$N emission factor ($^{15}$NO$_{Xi}$/$^{14}$NO$_{xi}$) calculated by:

$$^{15}R_{NO_x}(i) = \left(\frac{\delta^{15}N_{NO_x(i)}}{1000} + 1\right) \times 0.0036 \qquad \text{Eq. (3)}$$

$\delta^{15}$N$_{NOx(i)}$ is the $\delta^{15}$N value of some NO$_x$ source ($i$ = area, biogenic, mobile, and point) and 0.0036 is the $^{15}$N/$^{14}$N of air N$_2$, the reference point for $\delta^{15}$N values.

Annual NO$_x$ emissions for 2002 were obtained from the NEI at the county-level and were converted into hourly emissions on a 12 km x 12 km grid as previously published (Spak, Holloway, & Stone, 2007). The modeling domain includes latitudes between 37 º N and 45 º N, and longitudes between 98º W and 78º W, which fully covers the Midwestern US (Fig. 2, in yellow). SMOKE categorizes NO$_x$ emissions into four "processing categories": Biogenic, Mobile, Point, and Area (Table 1). The choice of the 2002 version of NEI is, in part, arbitrary. However, to compare the model predicted $\delta^{15}$N values with observations, it requires the emission inventory to be relevant to the same timeframe as the $\delta^{15}$N measurements of the NO$_y$. The data sets we compare to the model (discussed below) span from 2002 to 2009, thus the 2002 inventory is more relevant than later inventories (2014 onward). The county-level annual $^{14}$NO$_x$ emission for the Midwestern US from NEI was converted to the dataset with hourly $^{14}$NO$_x$ emissions. Livestock waste and off-road vehicles classified as area sources and each county was gridded evenly. Power plants are regarded as the point source and are located in grids corresponding to their latitudes and longitudes. On-road vehicles were regarded as the mobile source by SMOKE estimated by MOBILE model (see SA). The soil NO$_x$ produced by microbial nitrification and denitrification is classified as biogenic NO$_x$ emission and was estimated by BEIS model (see SA).

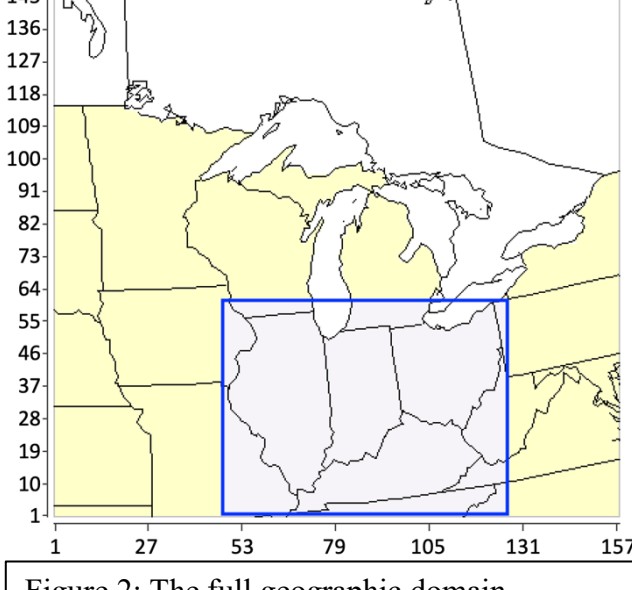

Figure 2: The full geographic domain (yellow) and extracted domain (light purple) for the study.

| SMOKE Category | NEI Sector | $\delta^{15}$N-NO$_x$ (‰) range | $\delta^{15}$N-NO$_x$ (‰) this study |
|---|---|---|---|
| Biogenic | Soil | -59.8 ~ -14.0 | -34.3 (Felix & Elliott, 2014) |
| Area | Livestock Waste | -29 ~ -8.5 | -18.8 (Felix & Elliott, 2014) |
| | Off-road Gasoline | -21.1 ~ 8.5 | -11.5 (Walters et al., 2015b) |
| | Off-road Diesel | | -10.5 (Walters et al., 2015b) |
| Mobile | On-road Gasoline | -28.1 ~ 17 | -2.7 (Walters et al., 2015b) |
| | On-road Diesel | | -2.5 (Walters et al., 2015b) |
| Point | Coal-fired Fossil Fuel Combustion | -19.7 ~ 25.6 | 15 (Felix et al., 2012) |
| | Natural Gas Fossil Fuel Combustion | | -16.5 (Walters et al., 2015) |

Table 1: The $\delta^{15}$N values (in ‰) for NO$_x$ emission sources based on SMOKE processing category and NEI sector

2.1.1 Biogenic $^{15}$NO$_x$ emissions

The NO$_x$ emission from the soil (Biogenic) was modeled in SMOKE using standard techniques (details in SA) and the $\delta^{15}$N values of biogenic NO$_x$ were taken from previous studies. Li & Wang (2008) measured the NO$_x$ fluxes using dynamic flow chambers for 2 to 13 days after cropland soil was fertilized by either urea (n=9) or ammonium bicarbonate (n=9), and the $\delta^{15}$N values of NO$_x$ ranged from -48.9 ‰ to -19.8 ‰. Felix & Elliott (2014) used passive samplers to collect NO$_2$ in a cornfield for 20 days, before and after fertilizer application. The $\delta^{15}$N values of

NO$_x$ emissions from these measurements range from -30.8 ‰ to -26.5 ‰. Using a similar methodology, Miller et al. (2018) collected NO$_2$ between May and June finding δ$^{15}$N ranging from -44.2 ‰ to -14.0 ‰ (n=37). Yu & Elliott (2017) measured -59.8 ‰ to -23.4 ‰ in 15 samples from soil plots in a fallow field 2 weeks after the precipitation. Based on these studies we adopted an average δ$^{15}$N value for NO$_x$ emissions from the soil of -34.3 ‰ (Li & Wang, 2008; Felix & Elliott, 2014; Yu & Elliott, 2017; Miller et al., 2018).

2.1.2 Mobile $^{15}$NO$_x$ emissions

The SMOKE NO$_x$ emission from on-road vehicles used standard methods (details in SA) and used δ$^{15}$N values from prior studies (Moore, 1977; Heaton, 1990; Ammann et al., 1999; Pearson et al., 2000; Savard et al., 2009; Redling et al., 2013; Felix & Elliott, 2014; Fibiger, 2014; Walters et al., 2015a, 2015b). We have excluded studies that infer NO$_x$ δ$^{15}$N by measuring plant proxies or passive sampling in the environment (Ammann et al., 1999; Pearson et al.,2000; Savard et al. 2009; Redling et al., 2013; Felix & Elliott, 2014). This is because of equilibrium and kinetic isotope effects that can occur as NO$_x$ reacts in the atmosphere to form NO$_y$, prior to NO$_x$ deposition. In addition, the role vegetation plays in NO$_x$ removal and atmospheric processes that mix the δ$^{15}$N of emission with the surroundings can also alter the δ$^{15}$N from the mobile source. Instead, we estimated the δ$^{15}$N value of NO$_x$ emissions from vehicles only using studies that directly measured tailpipe NO$_x$ emissions. Moore (1977) and Heaton (1990) collected tailpipe NO$_x$ spanning -13 ‰ to 2 ‰, with an average of -7.5 ± 4.7 ‰. Neither Heaton nor Moore noted whether these 6 vehicles were equipped with any catalytic NO$_x$ reduction technology, but it is unlikely since the late 1970 and 80's s vehicles were seldomly equipped with catalytic NO$_x$ reduction technology. Fibiger (2014) measured 5 samples of NO$_x$ from diesel engines without SCR emitted into a smog chamber, the δ$^{15}$N values range from -19.2 ‰ to -16.7 ‰ (±0.97 ‰). The most comprehensive studies on vehicle NO$_x$ δ$^{15}$N values are by Walters et al. (2015a, 2015 b) who measured gas and diesel vehicles separately, including those with and without three-way catalytic converter (TCC) and SCR technology. They also measured on-road and off-road vehicles separately. This research showed that the δ$^{15}$N of NO$_x$ for vehicles without SCR or when SCR was not functioning was negative, at around -15‰. As SCRs warmed and became efficient at reducing NO$_x$ the δ$^{15}$N value became less negative and even went positive. The measurements showed that the δ$^{15}$N values of NO$_x$ emitted by gasoline on-road vehicles averages at -2.5 ± 1.5 ‰, and on-road diesel ranged from -5 ‰ to 0 ‰.

The emission rate of $^{15}$NO$_x$ from the mobile source was determined by Eq. 4 grid by grid, according to the contributions from on-road gasoline vehicles and on-road diesel vehicles, as well as their corresponding δ$^{15}$N values of these two types of vehicles grid by grid. NO$_x$ emissions from off-road vehicles are regarded as area sources in SMOKE, which were processed over each county. In contrast, NO$_x$ emissions from on-road vehicles are regarded as the mobile source in SMOKE, which will be processed along each highway. Each grid emission rate of $^{15}$NO$_x$ was assigned based on the contributions from gasoline and diesel vehicles, as well as the relative δ$^{15}$N values. The δ$^{15}$N of on-road gasoline vehicles (-2.7 ± 0.8 ‰) was based on the average of the vehicle travel time within each region with the same zip code (Walters et al., 2015b).

$$^{15}NO_x \,(mobile) = \left(\frac{\delta^{15}N_{NO_x\,(on-road\,gas)}}{1000} + 1\right) \times 0.0036 \times {}^{14}NO_x\,(on-road\,gas)$$
$$+ \left(\frac{\delta^{15}N_{NO_x\,(on-road\,diesel)}}{1000} + 1\right) \times 0.0036 \times {}^{14}NO_x\,(on-road\,diesel) \quad \text{Eq. (4)}$$
$$\text{Where } \delta^{15}N_{NO_x\,(on-road\,gas)} = -12.35 + 3.02 \times \ln(t + 0.455)$$

2.1.3 Point source $^{15}NO_x$ emissions

$NO_x$ point sources are large anthropogenic $NO_x$ emitters located at a fixed, stationary position such as EGUs (electric generating units). Fugitive dust does not significantly contribute to point $NO_x$ emissions, so our inventory focused only on power plants (Houyoux, 2005). Power plants were separated into two different types: EGU and Non-EGU (e.g. commercial and industrial combustions). The $\delta^{15}N$ value of $NO_x$ emitted from power plants have been estimated to vary from -19.7 ‰ to 25.6 ‰ (Heaton, 1987; Heaton, 1990; Snape, 2003; Felix et al., 2012; Felix et al., 2015; Walters et al., 2015b; Savard et al., 2017). We have ignored studies that measured $\delta^{15}N$ of $NO_3^-$ or $HNO_3$ from EGUs (Felix et al., 2015, Savard et al., 2017) and instead, only consider those studies that directly measured $\delta^{15}N$ of $NO_x$. Heaton (1990) collected 5 samples from the different coal-fired power stations finding $NO_x$ from 6 ‰ to 13 ‰, with a standard deviation of 2.9 ‰. Snape (2003) measured $\delta^{15}N$ values of 36 samples from power plants using three different types of coals in combustion chars in a drop tube reactor, with values ranging from 2.1 ‰ to 7.2 ‰, with a standard deviation of 1.37 ‰. The most comprehensive study on coal-fired power plants' $NO_x$ values was by Felix et al. (2012). They measured the $\delta^{15}N$ values of $NO_x$ emission from the coal-fired power stations with and without different emission control technologies. 16 coal-fired power plants with SCR, 3 coal-fired power plants with SNCR, 15 coal-fired power plants with OFA/LNB, and 8 coal-fired power plants without emission control technology were measured. The $\delta^{15}N$ values of $NO_x$ emissions from these 42 measurements range from 9 ‰ to 25.6 ‰, with a standard deviation of 4.51 ‰. The $NO_x$ $\delta^{15}N$ values when different emission control technologies were used varied: the $\delta^{15}N$ values of $NO_x$ emissions from coal-fired power plants with SCR range from 15.5 ‰ to 25.6 ‰, those with SNCR ranged from 13.6 ‰ to 15.1 ‰, and those with OFA/LNB ranged from 9.0 ‰ to 12.6 ‰. The $\delta^{15}N$ values of $NO_x$ emissions from coal-fired power plants without emission control technology range from 9.6 ‰ to 11.7 ‰, with a standard deviation of 0.79 ‰. According to Xing et al. (2013), about half of the coal-fired power plants in the United States are equipped with SCR. Thus, we assume 15 ‰ for the $NO_x$ emissions from coal-fired power plants, which is the average between SCR and other emission control technologies.

The most comprehensive study on natural gas-fired $NO_x$ values (Walters et al. 2015) collected 12 flue samples on the rooftop of a house from the ventilation pipe of a natural gas low-$NO_x$ burner residential furnace without $NO_x$ emission control technology. The measurement showed that the $\delta^{15}N$ values of $NO_x$ emitted by natural gas power plants average $-16.5 \pm 1.7$ ‰, which we used for the $NO_x$ emission from natural gas power plants. The reason for using these values is because they were measurements taken directly from the exhaust pipes, rather than inferring from downwind area or from rain samples, emitted by natural gas power plants, and included power plants with and without SCR technology. The latitude, longitude, and point sources characteristics (EGU and non-EGU, coal-fired or natural gas-fired, implementation of emission control technology) of each power plant was obtained from the US Energy Information Administration (2017). The power plants were assigned grids by their latitudes and longitudes, and the $\delta^{15}N$ values were assigned to these grids based on their emission characteristics, before determining the emission rate of $^{15}NO_x$ from point source using Eq. (2) and (3).

2.1.4 Area source $^{15}NO_x$ emissions

Area $NO_x$ (details in SA) $\delta^{15}N$ values were based on the assumption that livestock waste and off-road vehicles (utility vehicles for agricultural and residential purposes) accounted for total area sources. Livestock waste $NO_x$ $\delta^{15}N$ values were taken from Felix & Elliott (2014) since it is currently the only study about the $\delta^{15}N$ value of $NO_x$ livestock waste emissions. They placed a

passive sampler with ventilation fans in an open-air and closed room in barns of cows and turkeys, respectively. The $\delta^{15}N$ values of $NO_x$ emissions from these measurements range from -29 ‰ to -8.5 ‰. Among these samples, the $\delta^{15}N$ of $NO_x$ emissions from turkey waste averages at -8.5 ‰, the $\delta^{15}N$ of $NO_x$ emissions from cow waste averages at -24.7 ‰. We used -18.8 ‰ as the values of $\delta^{15}N$ values for $NO_x$ emissions from livestock waste, which is the weighted average of the $\delta^{15}N$ of $NO_x$ from turkey waste and cow waste emissions. We used the $\delta^{15}N$ values from Walters et al. (2015b) to estimate the $\delta^{15}N$ value of $NO_x$ emissions from the off-road vehicle since it is the latest in-depth study that measured the $\delta^{15}N$ value of $NO_x$ specifically from the off-road vehicle. They collected 45 samples from the tailpipe of 9 different off-road vehicles (gasoline and diesel) with and without SCR, and before and after the sufficient engine warm-up times. The measurement showed that the $\delta^{15}N$ values of $NO_x$ emitted by gasoline-powered off-road vehicles averaged -11.5 ± 2.7 ‰, diesel off-road vehicles without SCR averaged -19 ‰ ± 2 ‰, and diesel off-road vehicles with SCR averaged -2 ‰ ± 8 ‰. The emission rate of $^{15}NO_x$ from area source was determined by Eq. 5 grid by grid, according to the contributions from waste, off-road gasoline vehicle, and off-road diesel vehicle, as well as their corresponding $\delta^{15}N$ values based on previous researches.

$$^{15}NO_x\ (area) = \left(\frac{\delta^{15}N_{NO_x\ (waste)}}{1000} + 1\right) \times 0.0036 \times {}^{14}NO_x\ (waste)$$

$$+ \left(\frac{\delta^{15}N_{NO_x\ (off-road\ gas)}}{1000} + 1\right) \times 0.0036 \times {}^{14}NO_x\ (off-road\ gas)$$

$$+ \left(\frac{\delta^{15}N_{NO_x\ (off-road\ diesel)}}{1000} + 1\right) \times 0.0036 \times {}^{14}NO_x\ (off-road\ diesel) \qquad \text{Eq. (5)}$$

The $^{15}NO_x$ emission data files of each SMOKE processing category was incorporated into the final dataset based on the $\delta^{15}N$ values from previous research (Table 1) and Eq. (2-5).

$$\delta^{15}N_{NO_x\ (total)} = \left(\frac{\frac{^{15}NO_x\ (area) + {}^{15}NO_x\ (biog) + {}^{15}NO_x\ (mobile) + {}^{15}NO_x\ (point)}{^{14}NO_x\ (area) + {}^{14}NO_x\ (biog) + {}^{14}NO_x\ (mobile) + {}^{14}NO_x\ (point)}}{0.0036} - 1\right) \times 1000 \qquad \text{Eq. (6)}$$

## 2.2 Simulating atmospheric $\delta^{15}N(NO_x)$ in CMAQ

In order to investigate the role of mixing in the spatiotemporal distribution of $NO_x$ $\delta^{15}N$ values, CMAQ was used to simulate the meteorological transport effects (advection, eddy diffusion, etc). In this "emission + mixing" scenario grid specific $NO_x$ $\delta^{15}N$ values emitted blur as $NO_x$ mixes across the regional scale. This blurring will depend on grid emission strength and mixing vigor and is effectively treating $NO_x$ as a conservative tracer. The simulations used the 2002 National Emission Inventory (NEI), as well as 2002 and 2016 meteorological conditions respectively, in order to explore how meteorological conditions will impact the atmospheric $\delta^{15}N(NO_x)$. Simulations covering the full domain and extracted domain were conducted, in order to explore and eliminate the bias near the domain boundary.

In addition, CMAQ simulated the $\delta^{15}NO_x$ effect by $NO_x$ removal using enhanced deposition. These "emission + mixing + enhanced deposition" simulations were **not** imposing an isotope effect related to dry/wet deposition, rather they are an attempt to show how "lifetime chemistry" alters $NO_x$ $\delta^{15}N$ values by removing $NO_x$ before it can be transported significant distances. For example, in an "emission + mixing" scenario $NO_x$ from a high emission powerplant could travel across the domain altering regional $NO_x$ $\delta^{15}N$ as it mixes with other grids. By contrast, in the "emission + mixing + enhanced deposition" scenario most of that $NO_x$ would be removed near the power plant,

effectively constricting its $\delta^{15}N$ influence. This enhanced deposition effect was simulated by disabling the chemistry module in CMAQ and enhancing the $NO_x$ dry deposition rates (discussed in 2.2.3). This has an added advantage in that the deposited $NO_x$ $\delta^{15}N$ should be similar to the $NO_3^-$ $\delta^{15}N$, which is not being generated in this model. We emphasize that in this model the isotope effects associated with the photochemical transformation of $NO_x$ into $HNO_3$ (and other higher N oxides) and deposition are ignored and will be addressed in the forthcoming paper.

2.2.1 Meteorology input dataset

To explore the impact of atmospheric processes, the meteorology input datasets for the years 2002 and 2016 were prepared and compared. The CMAQ CTM (CCTM) used the NARR (North American Regional Reanalysis) and NAM (North American Mesoscale Forecast System) to convert the weather observations (every 3 hours for NARR, every 6 hours for NAM Analyses) into gridded meteorological elements, such as temperature, wind field, and precipitation, with the horizontal resolution of 12 km, and 34 vertical layers, with the thickness, increases with height, from 50 m near the surface to 600 m near the 50 mb pressure level. These were used to generate the gridded meteorology files on an hourly basis, using the Weather Research and Forecasting Model (WRF). To maintain consistency between the $NO_x$ emission dataset and the meteorology, the same coordinate system, spatial domain, and grid size used in the SMOKE model were used in the WRF simulation. The WRF outputs were used to prepare the CMAQ-ready meteorology input dataset using CMAQ's MCIP (the Meteorology-Chemistry Interface Processor; see SA for details). In these emission-only simulations, the deposition of $NO_x$ was effectively set to zero. This was accomplished by defining YO $=^{14}NO$ and $YO_2 = {}^{14}NO_2$ (in addition to ZO $=^{15}NO$ and $ZO_2 = {}^{15}NO_2$) and setting their VDs (deposition velocities) to 0.001 (since setting them to zero collapses the simulation) in the namelist for the gas-phase species (GC_cb6r3_ae6_aq.nml).

2.2.2 Initial condition and boundary condition for the simulation

The meteorological fields generated by MCIP were used as the inputs for Initial Conditions Processor (ICON) and Boundary Conditions Processor (BCON), used for running CCTM of CMAQ. The ICON program prepares the initial chemical/isotopic concentrations in each of the 3D grid cells for use in the initial time step of the CCTM simulation. The BCON program prepares the chemical/isotopic boundary condition throughout the CCTM simulation. The CMAQ default ICON and BCON for a clean atmosphere were used, which had $NO_x < 0.25$ ppb. The $^{15}NO_x$ were added to the outputs of ICON and BCON, with the concentration equal to $0.0036[^{14}NO_x]$, which assumes $\delta^{15}N = 0$ at the initial time step and outside the domain of the simulation.

2.2.3 The role of deposition and chemical transformation of $NO_x$

The deposition rates $^{14}NO_x$ and $^{15}NO_x$ were varied to assess their role in the spatiotemporal distribution of $NO_x$ $\delta^{15}N$ value and to emulate photooxidation of $NO_x$. In these "emission + mixing + enhanced deposition" simulations, the molecular mass of Y and Z were set equal (14) to ensure no isotope effect was induced by dry deposition, since the equations for dry deposition have a mass term in the diffusion coefficient calculation. The $^{15}NO_x/NO_x$ deposition rates were amplified by first magnifying it to 20 times normal (14 kg/hectare/yr) and testing for the change in $NO_x$ concentration relative to the normal deposition rate. Multiple tuning trials were conducted until the e-folding time (lifetime) of $NO_x$ in the atmosphere across the domain averaged about 1 day. This is a typical average $NO_x$ lifetime for a combination of urban, suburban, and rural environments (Laughner & Cohen, 2019).  This approach is limited since $NO_x$ lifetime varies

depending on oxidation capacity, with urban $NO_x$ lifetimes (~2-11 hours) being significantly shorter than in rural conditions (Fang et al., 2021). This limitation will be resolved once $^{15}N$ is included in the gas and aerosol chemistry modules to future versions of CMAQ.

2.2.4 The simulation over the extracted domain

As mentioned in section 2.2.4, atmospheric $NO_x$ $\delta^{15}N = 0‰$ for initial condition and boundary condition. As a result, the bias occurs near the border of the research area, mainly under the following two circumstances. Firstly, when the air mass transports out of the research area (Fig. S1). Due to the lack of the emission dataset, Canada is considered an "emission-free zone" for this research. As a result, the atmospheric $NO_x$ is diluted, which impacts its $\delta^{15}N$ values, especially for those with extreme $\delta^{15}N$ values ($\delta^{15}N < -15‰$ or $\delta^{15}N > 5‰$). Secondly, the air mass with $\delta^{15}N(NO_x) = 0$ transports from the "emission-free zone" to the research area (Fig. S2), the atmospheric $\delta^{15}N(NO_x)$ is flattened. Therefore, to avoid the bias near the border, the extracted domain that only covers Indiana, Illinois, Ohio, and Kentucky was determined (Fig. 2, in light purple), where the measurements of $\delta^{15}N$ values at NADP sites are available (Mase, 2010; Riha, 2013). The boundary condition for the simulation over the extracted domain is based on the CCTM output of the full-domain simulation (BCON code available on Zenodo.org (10.5281/zenodo.4311986)).

3. Results and Discussion

3.1 Simulated spatial variability of $NO_x$ emission rates

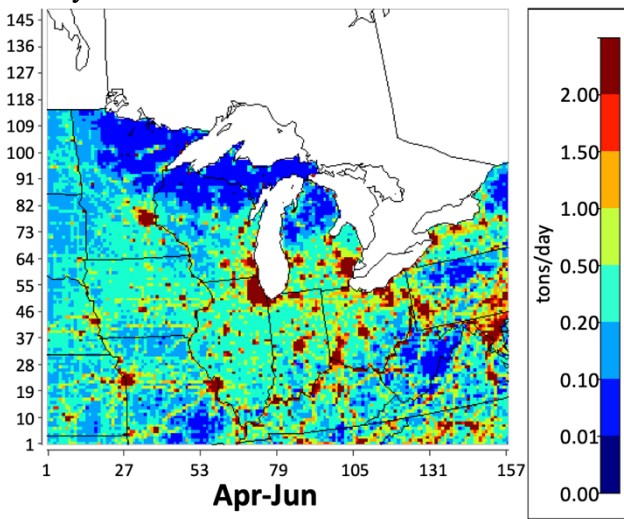

Figure 3: Total $NO_x$ emission in the Midwest between April and June in tons/day. High $NO_x$ emissions are associated with major urban areas such as Chicago, Detroit, Minneapolis-St Paul, Kansas City, St. Louis, Indianapolis, and Louisville.

We first examine the spatial heterogeneity of the $NO_x$ emission rate for a single time period to illustrate the overall pattern of $NO_x$ emission over the domain (Fig. 3). This is because the $\delta^{15}N$ value of total $NO_x$ emission is determined by the fraction of each $NO_x$ source (Eq. 6), which in turn is a function of their emission rates. Since our $NO_x$ emissions are gridded by SMOKE using the NEI, they are, by definition, corrected with respect to the NEI. However, a brief discussion of

the salient geographic distribution of $NO_x$ emissions and comparisons with other studies is warranted for completeness and as a backdrop for the discussion of $NO_x$ fractions and resulting $\delta^{15}N$ values. We have arbitrarily chosen to sum the $NO_x$ emissions during the April to June time period for this discussion (Fig. 3).

The seasonal average $NO_x$ emissions within the geographic domain during April to June range from less than 0.01 tons N/day to more than 15 tons N/day, with the seasonal grid average of 0.904 tons/day. This average agrees well with estimates in previous studies for the United States, which were between 0.81 and 1.02 tons/day (Dignon & Hameed, 1989; Farrell et al., 1999; Selden et al., 1999; Xing et al, 2012). Within 75% of the geographic domain, the $NO_x$ emissions are relatively low, ranging from between 0 and 0.5 tons/day (Fig. S3). Geographically, these grids are located in rural areas some distance away from metropolitan areas and highways (Fig. 3). $NO_x$ emissions within about 20% of the grids is relatively moderate, ranging between 0.5 and 2.0 tons/day (Fig. S3). Geographically, these grids are mainly located along major highways and areas with medium population densities (Fig. 3). Urban centers comprise about 5% of the grids within the geographic domain and these have high $NO_x$ emissions rates, ranging between 2.0 and 15.0 tons/day (Fig. S3). The metropolitan area's average is 5.03 tons/day, which is nearly 14 times of the average emission rate over the rest of the grids within the geographic domain (0.37 tons/day) due to the high vehicle density associated with high population densities. The highest emissions rates are located within large cities (Fig. 3), such as Chicago, Detroit, Minneapolis-St Paul, Kansas City, St. Louis, Indianapolis, and Louisville, as well as the edge of the east coast metropolitan area (dark red). Summing the $NO_x$ emissions among the grids that encompass these major midwestern cities, yields city-level $NO_x$ emission rates that vary from 61.2 tons/day (Louisville, KY) to 634.1 tons/day (Chicago, IL). These city-level $NO_x$ emission rates (Table S4) agree well with estimates derived from the Ozone Monitoring Instrument (Lu et al., 2015). Grids containing power plants are the significant $NO_x$ hotspots within the geographic domain. These account for less than 1% of the grids, but the $NO_x$ emissions from a single grid that contains a power plant can be as high as 93.4 tons/day. Geographically, the power plants are mainly located along the Ohio River valley, near other water bodies, and often close to metropolitan areas (Fig. 3). The $NO_x$ emission rates of the major power plants within the Midwest simulated by SMOKE (Table S5) match well with the measurement from the Continuous Emission Monitoring System (CEMS) (de Foy et al., 2015; Duncan et al., 2013; Kim et al., 2009). The geographic distribution of grid-level annual $NO_x$ emission density in our simulation also agrees with the county-level annual $NO_x$ emission density discussed in the 2002 NEI booklet (Fig. S4; USEPA, 2018b).

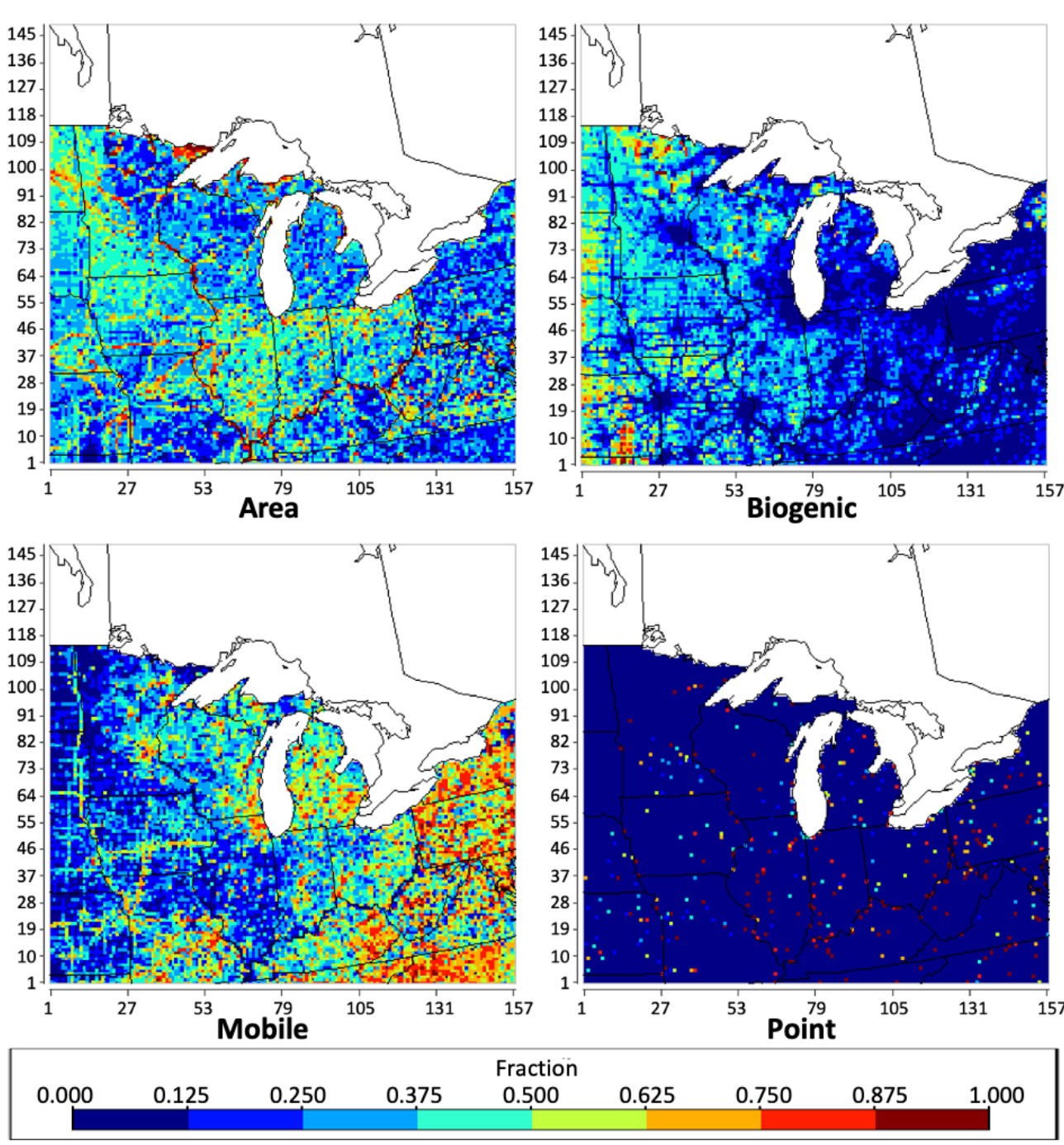

Figure 4: The geographical distribution of the fraction of NO$_x$ emission from each SMOKE processing category (area, biogenic, mobile, point) over each grid throughout the Midwest between April and June based on NEI-2002.

3

We next examine the spatial heterogeneity of the $NO_x$ source fractions (Fig. 4) for the same
time period (April to June). The $NO_x$ fraction ($f$) is defined as the amount of $NO_x$ from a source
category normalized to total $NO_x$ ($f_s$ =$NO_x$(source)/$NO_x$(total).   Since the $\delta^{15}NO_x$ is determined
by the $NO_x$ emission fractions within each grid it is important to understand where in the domain
these fractions differ and why. The area sources, which mainly consist of off-road vehicles,
agriculture production, residential combustion, as well as the industrial processes, which are
individually too low in magnitude to report as point sources, are fairly uniform in their distribution
across the domain.
The SMOKE simulation shows that $NO_x$ emissions from area sources contribute an average
$NO_x$ emission fraction ($f_{area}$) of 0.271 for total $NO_x$ emission and 0.290 for anthropogenic $NO_x$
emission within the Midwest from April to June. The fractions of $NO_x$ emission from area sources
show a clear spatial variation and range from 0.125 to 0.5 over about 75% of the grids (Fig. S5).
Geographically, the grids with relatively higher $f_{area}$ are located in the rural area away from
highways, especially in the states of Indiana, Illinois, Iowa, Minnesota, and Ohio, where
agricultural is the most common land use classification. In the states of Wisconsin and Missouri,
the $f_{area}$ is slightly lower due to the higher fraction of $NO_x$ emission from biogenic sources ($f_{biog}$).
In the states of Pennsylvania and Michigan, the $f_{area}$ is slightly lower due to the higher fraction of
$NO_x$ emission from mobile sources ($f_{mobile}$). In addition, the grids with $f_{area}$ greater than 0.75 are
mainly located along the Mississippi River and Ohio River, due to wastewater discharge.
The fraction of biogenic $NO_x$ ($f_{biog}$) that are predominately by-products of microbial
nitrification and denitrification occurring in soil, shows the clear spatial variation and is highest
(from April to June) in the western portion of the domain (Fig. 4). The average fraction of biogenic
$NO_x$ emission within the Midwest from April to June and is 0.065, which is less than 0.5 in more
than 90% of the grids within the geographic domain (Fig. S5). Geographically, the grids with
relatively high $f_{biog}$ are located in the western regions of the Midwest, away from cities and
highways, in the states of Minnesota, Iowa, Missouri, Wisconsin, and Illinois, where the density
of agricultural acreage and natural vegetation is higher than other states. Furthermore, within
regions with higher $f_{biog}$, the obvious low $f_{biog}$ values occur in the megacities and along the
highways, which agrees well with the land-use related to the biogenic emission.
The SMOKE simulation shows that the $NO_x$ emissions from mobile sources contribute to the
fraction ($f_{mobile}$) of 0.325 for total $NO_x$ emission and 0.347 for anthropogenic $NO_x$ emission within
the Midwest from April to June. The $f_{mobile}$ shows a clear spatial variation, with relatively higher
$f_{mobile}$ are located in major metropolitan regions and along the highways, where vehicles have the
highest density. In addition, within the states with lower $f_{mobile}$, the obvious high $f_{mobile}$ values occur
in the megacities and along the highways, which agrees well with the vehicle activities (US Census
Bureau, n.d.). The value of $f_{mobile}$ within the geographic domain distributes evenly on the histogram
(Fig. S5).
The point sources consist mainly of EGUs, as well as commercial and industrial processes
involving combustion. Based on the SMOKE simulation, the $NO_x$ emission from point sources
contributes to the fraction ($f_{point}$) of 0.339 for total $NO_x$ emission and 0.363 for anthropogenic $NO_x$
emission within the Midwest from April to June. The fractions of $NO_x$ emission from the point
source over each grid cell within the geographic domain show a clear spatial variation.
Geographically, the $NO_x$ emission from point sources is dominant at the grids, where the power
plants are located, mainly along the Ohio River valley and near other water bodies close to
metropolitan areas. The point sources have no contribution to the $NO_x$ emission among about 96%
of the grids within the geographic domain. The rest of the 4% of the grids within the geographic
domain are the locations of power plants. About 1/4 of the power plants are not at the same grids
as highways, thus these grids have a fraction of at least 0.9 $NO_x$ emission from point sources.
Whereas the other 3/4 of the power plants share the same grids with highways/cities, thus the point
sources become relatively less dominant, due to the dilution by the $NO_x$ emission from mobile
sources.
## 3.2  Simulated spatial variability in $\delta^{15}NO_x$

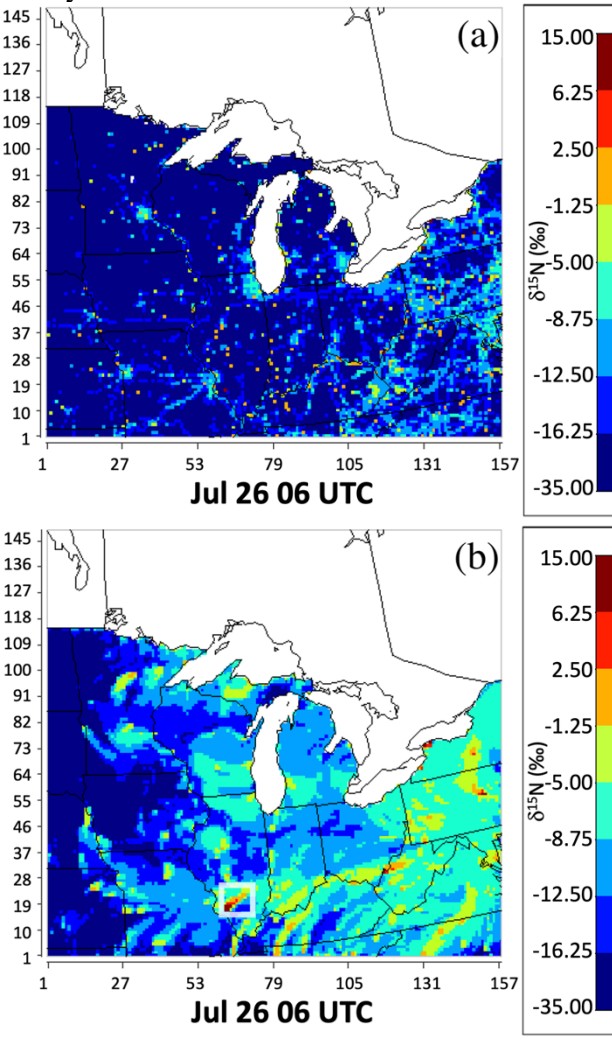

Figure 5: The $\delta^{15}N$ values of $NO_x$ emission, (a: "no transport" scenario) and the $\delta^{15}N$ values of atmospheric $NO_x$ based on NEI-2002 and 2016 meteorology (b: "with transport" scenario), at 06 UTC on July 26, are presented by color in each grid. The warmer the color, the higher $\delta^{15}N$ values of atmospheric $NO_x$.

9       Using these $NO_x$ emission source fractions in each grid, the $\delta^{15}N$ values of $NO_x$ were
simulated. Here, the spatial heterogeneity of $\delta^{15}N$ values of $NO_x$ for a single time period is
discussed. The "emission only" simulation of $NO_x$ $\delta^{15}N$ values (at 06 UTC on July 26) ranged
from -34.3‰ to 14.9‰ (Fig. 5a). The majority of the grids within the domain have $NO_x$ $\delta^{15}N$
values lower than -16.3‰. These low $\delta^{15}N$ values across most of the domain are due to the $\delta^{15}N$
of -34.3‰ for biogenic $NO_x$ emission sources in sparsely populated areas where intensive
agriculture dominates the land use (Fig. 5a). The $NO_x$ $\delta^{15}N$ values for grids within big cities mainly
ranged between -8.75‰ and -5‰ due to the higher fraction of $NO_x$ emission from on-road vehicles
($\delta^{15}N$ = -2.7 ± 0.8‰), which also resolve major highways. The highest value of $\delta^{15}N$ occurs at the
grids, where the coal-fired EGUs (+15‰) and hybrid-fired EGUs are the dominant $NO_x$ source
(Fig. 5a).

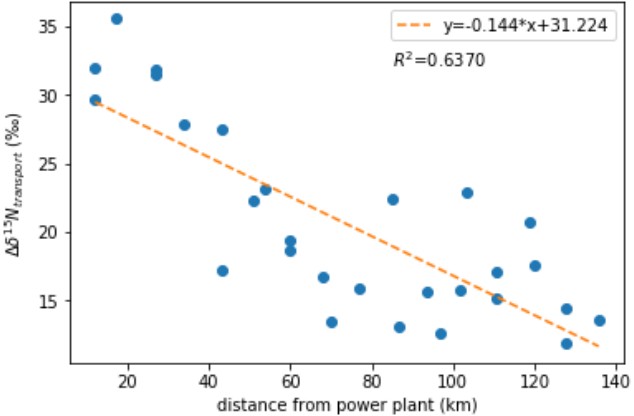

Figure 6: The $\Delta\delta^{15}N_{transport}$ along the plume (colored in dark red to orange inside the white box on Fig. 5b) over the distance from the power plant Baldwin Energy Complex (located at southwestern border of Illinois).

The effect of atmospheric mixing on the $\delta^{15}NO_x$ spatial distribution was then taken into
account by coupling the $^{15}NO_x$ emissions to the meteorology simulation. There are significant
differences between $\delta^{15}NO_x$ values in the "emission only" (Fig. 5a) and the "emission + transport"
(Fig. 5b) simulations. For example, under the "emission only" scenario (Fig. 5a) the map of $\delta^{15}NO_x$
values clearly shows the locations of big cities, major highways, and power plants, but these
features are much less obvious in the "emission + transport" (Fig. 5b) simulations. The isotopically
heavier $NO_x$ emission from big cities disperses to the surrounding rural areas so that the $\delta^{15}NO_x$
values in rural areas become elevated relative to the emission-only simulation. Similarly, the $NO_x$
emitted along major highways is transported to the surrounding grids, so that the atmospheric $NO_x$
at the grids around the major highways becomes isotopically heavier relative to the "emission only"
scenario. We define $\Delta\delta^{15}N_{transport}$ as the $\delta^{15}N$ difference between "emission only" and "emission +
transport" scenarios within the grids covered by the plume to quantify this effect (Fig. 6). The most
obvious and interesting example is the influence of grids containing coal-fired EGUs on the
surrounding region. For example, the southern Illinois' Baldwin Energy Complex (marked with a
transparent white box on Fig. 5b) that uses subbituminous coal and bituminous coal as its major
energy source. The $\Delta\delta^{15}N_{transport}$ in the regions is altered as a function of distance away from the
EGU. In this time snapshot (06 UTC on Jul 26), the northeastwards propagating plume of $NO_x$
emission from the EGU creates higher $\delta^{15}NO_x$ over 135 km away (Fig. 6). The domain average
$\delta^{15}N$ increases from -20.2‰ under the "emission only" scenario to -11.5‰ under the "emission +
transport" scenario. While "emission only" $\delta^{15}N$ pattern shows biogenic emission dominating the

spatial domain, in the "emission + transport" simulation anthropogenic emissions, becomes
dominant over most of the grids, especially for the grids located around major cities' power plants.

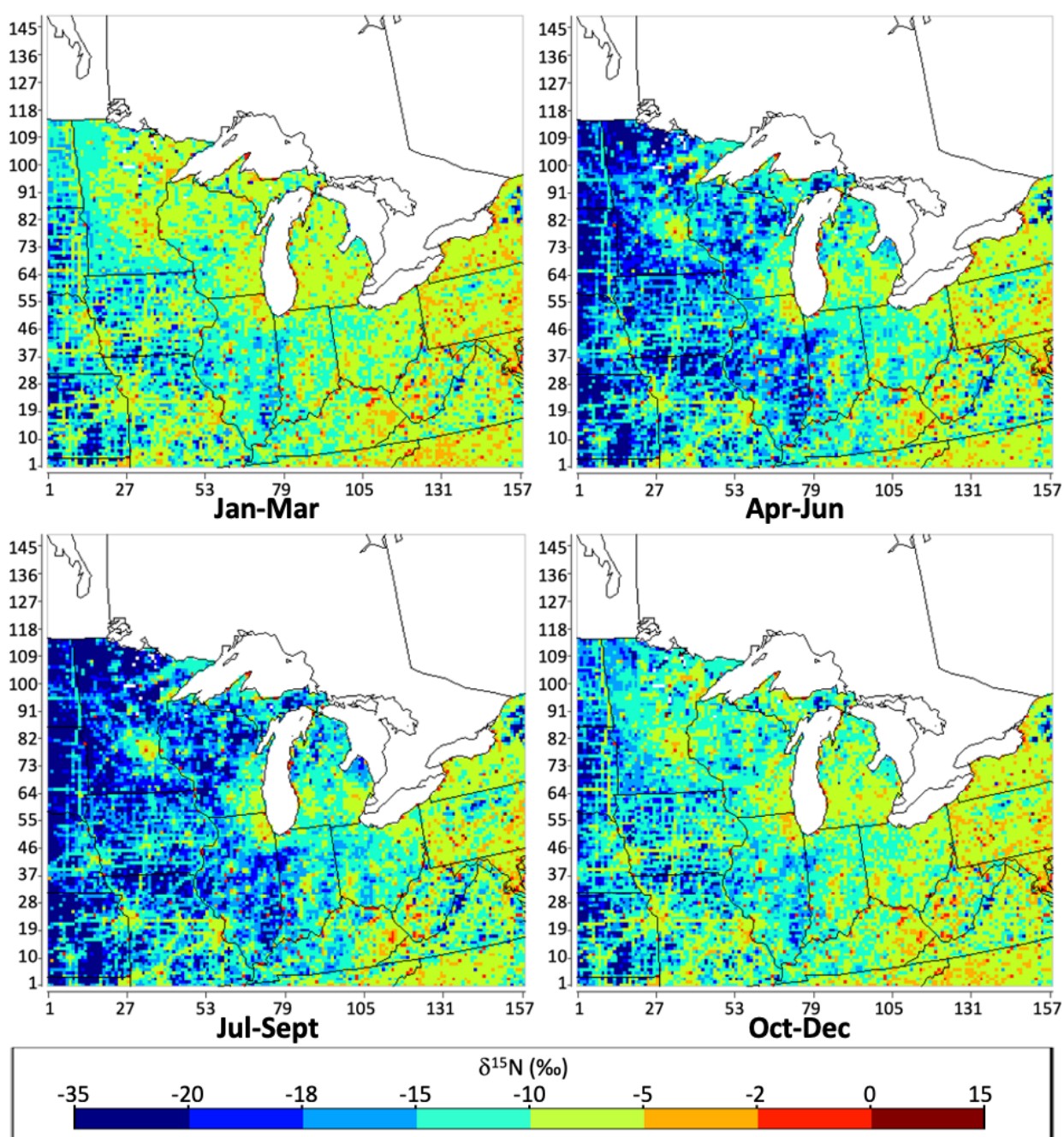

Figure 7: The geographical distribution of the $\delta^{15}N$ value of total $NO_x$ emissions in each season (Winter: Jan-Mar; Spring: Apr-Jun; Summer: Jul-Sep; Fall: Oct-Dec) in per mil (‰) throughout the Midwest simulated by SMOKE, based on NEI-2002.

3.3 Seasonal variation in $\delta^{15}NO_x$

2        We next examine the temporal heterogeneity of $\delta^{15}NO_x$ values over the domain for "emission
only) and interpret them in terms of changes in $NO_x$ emission fractions as a function of time. The
predicted $\delta^{15}N$ value of total $NO_x$ emissions in the Midwest during each season shows a significant
temporal variation (Fig. 7). The $\delta^{15}NO_x$ ranged from -35 ‰ to 15 ‰, with the annual average over
the Midwest at -6.15 ‰. The maps for different seasons show the obvious changes in $\delta^{15}N$ values
over western regions of the Midwest, from green ($\delta^{15}N$ = -15 ~ -5 ‰) to dark blue (-35 ~ -15 ‰)
during the month from April to October.

9        In order to qualitatively analyze the changes in $\delta^{15}NO_x$ among each season, the value of each
grid (Fig. 7) were organized into the histograms (Fig. S6), in order to show the percentage of the
grid in each color scheme. The grids with $\delta^{15}NO_x$ between -35‰ and -18‰ increase dramatically
from less than 10% during fall (Oct-Dec) and winter (Jan-Mar) to more than 20% during spring
(Apr-Jun) and summer (Jul-Sep). The grids with $\delta^{15}NO_x$ between -18‰ and -2‰ decrease from
around 90% during fall and winter to around 75% during spring and summer. In addition, the
distribution of $\delta^{15}NO_x$ shifts to lower values during spring and summer.

16       The significant temporal variation in the $\delta^{15}N$ value of total $NO_x$ during different seasons can
be quantitatively explained by changing fractions of $NO_x$ emission from the biogenic source in
any grid (Fig. S7) using Eq. (6). Unlike other $NO_x$ emission sources, the fraction of $NO_x$ emission
from biogenic sources changes significantly among each season within the geographic domain,
especially over the rural areas of the states of Minnesota, Iowa, Missouri, Wisconsin, Illinois,
Indiana, Kentucky, Michigan, and Ohio (Fig. S7). The fraction of $NO_x$ emission from biogenic
sources over these areas increases from less than 0.25 to more than 0.50 during the months of April
to October, which is the growing season. During this period, the surface sunlight hours,
temperature, and precipitation are relatively higher and as a result, the canopy coverage of the
plants becomes higher, which leads to the increase of the $NO_x$ emission from biogenic sources
(Pierce, 2001; Vukovich & Pierce, 2002; Schwede et al., 2005; Pouliot & Pierce, 2009; USEPA,
2018a). Besides this, the fertilizer application during this period is also responsible for the increase
in soil $NO_x$ emission (Li & Wang, 2008; Felix & Elliott, 2014).

29       In order to qualitatively analyze the changes in the fraction of $NO_x$ emission from biogenic
sources among each season, the distributions of the fractions among the same cut-offs as the maps
on Fig. S7 were shown in the histograms (Fig. S8). Comparing the distributions of the fractions of
$NO_x$ emission from biogenic sources among the histograms for each season, the effects from the
increasing of biogenic $NO_x$ emission during the growing season of plants are clearly shown. In
general, the distribution of the fraction shifts to higher values during spring (Apr-Jun) and summer
(Jul-Sep), indicating the increase of biogenic emissions. As a result, the distribution of $\delta^{15}NO_x$
shifts to lower values during the same period (Fig. 7). The percentage of the grids with the fraction
of biogenic emission less than 0.125 decreases dramatically from more than 50% during fall (Oct-
Dec) and winter (Jan-Mar) to less than 35% during spring (Apr-Jun) and summer (Jul-Sep). As the
$NO_x$ emission from biogenic source becomes dominant, the percentage of the grids with $\delta^{15}NO_x$
between -35‰ and -18‰ increases, while the percentage of the grids with $\delta^{15}N(NO_x)$ between -
18‰ and -2‰ decreases, which sufficiently explains the trends shown on Fig. 7.

We then examine the temporal heterogeneity of atmospheric $\delta^{15}NO_x$ under the "emission + transport" scenario over the domain and interpret them in terms of changes in the propagation of $NO_x$ emission as a function of time. The predicted seasonal average $\delta^{15}NO_x$ in the Midwest shows significant variations (Fig. 8). On an annual basis, the $\delta^{15}NO_x$ values range from -19.2‰ to 11.6‰, with the annual average over the Midwest domain of -6.10‰. Compared with the seasonal $\delta^{15}NO_x$ under the "no transport" scenario, the $\delta^{15}NO_x$ under the "with transport" scenario has a similar overall average while narrower range, due to the transport and mixing of the air mass. This could be clearly shown on the map, of which the color scheme is smoother, comparing with the seasonal $\delta^{15}N(NO_x)$ under the "no transport" scenario (Fig. 7). The maps for different seasons show the obvious changes in $\delta^{15}N$ values over western regions of the Midwest, from -8.75 ~ -5‰ in Oct-Mar to -16.25 ~ -12.5‰ in Apr-Oct.

In addition to the variability of the $NO_x$ emission source, the significant temporal variation in the $\delta^{15}N$ value of atmospheric $NO_x$ during different seasons is controlled by the transport and mixing of the air mass, under the different meteorology conditions that vary by season. The PBL height is an effective indicator showing whether the pollutants are under the synoptic condition, which is favorable for the dispersion, mixing, and transport after being emitted into the atmosphere (Oke, 2002; Shu et al., 2017; Liao et al., 2018; Miao et al., 2019). In order to qualitatively analyze the changes in $\delta^{15}N$ values driven by atmospheric processes, the difference between the $\delta^{15}N$ value of atmospheric $NO_x$ under the "emission + transport" scenario and "emission only" scenario ($\Delta\delta^{15}N_{transport}$) on the seasonal basis were shown (Fig. S9). The seasonal $\Delta\delta^{15}N_{transport}$ values range from -21.9‰ to 31.2‰, with an average of 4.9‰. The overall pattern of the $\Delta\delta^{15}N_{transport}$ values shows that after the $NO_x$ being emitted into the atmosphere, it became isotopically heavier over the majority of the grids within the domain, and isotopically lighter over the grids that contain big cities, major highways, and power plants. This could be explained by the transport and dispersion of biogenic emissions and anthropogenic emission to the surrounding areas. Among the grids located in rural areas, where the biogenic emission dominates the $NO_x$ budget, the $\delta^{15}N$ values increases from around -30‰ to around -10‰, due to transport and dispersion of anthropogenic emission with relatively high emission rates from surrounding cities, highways, or power plants, which brings the isotopically heavier $NO_x$ into the grids. On the other hand, among the grids located in the urban area, highways, or power plants, where anthropogenic emission dominates the $NO_x$ budget, the changes in $\delta^{15}N$ values decrease is much less obvious, showing the $\Delta\delta^{15}N_{transport}$ values ranges between -5‰ and +5‰. This could be explained by the relatively high rates of anthropogenic emissions. Thus, the effects of the transport and dispersion of biogenic emissions from the surrounding rural area are minimal.

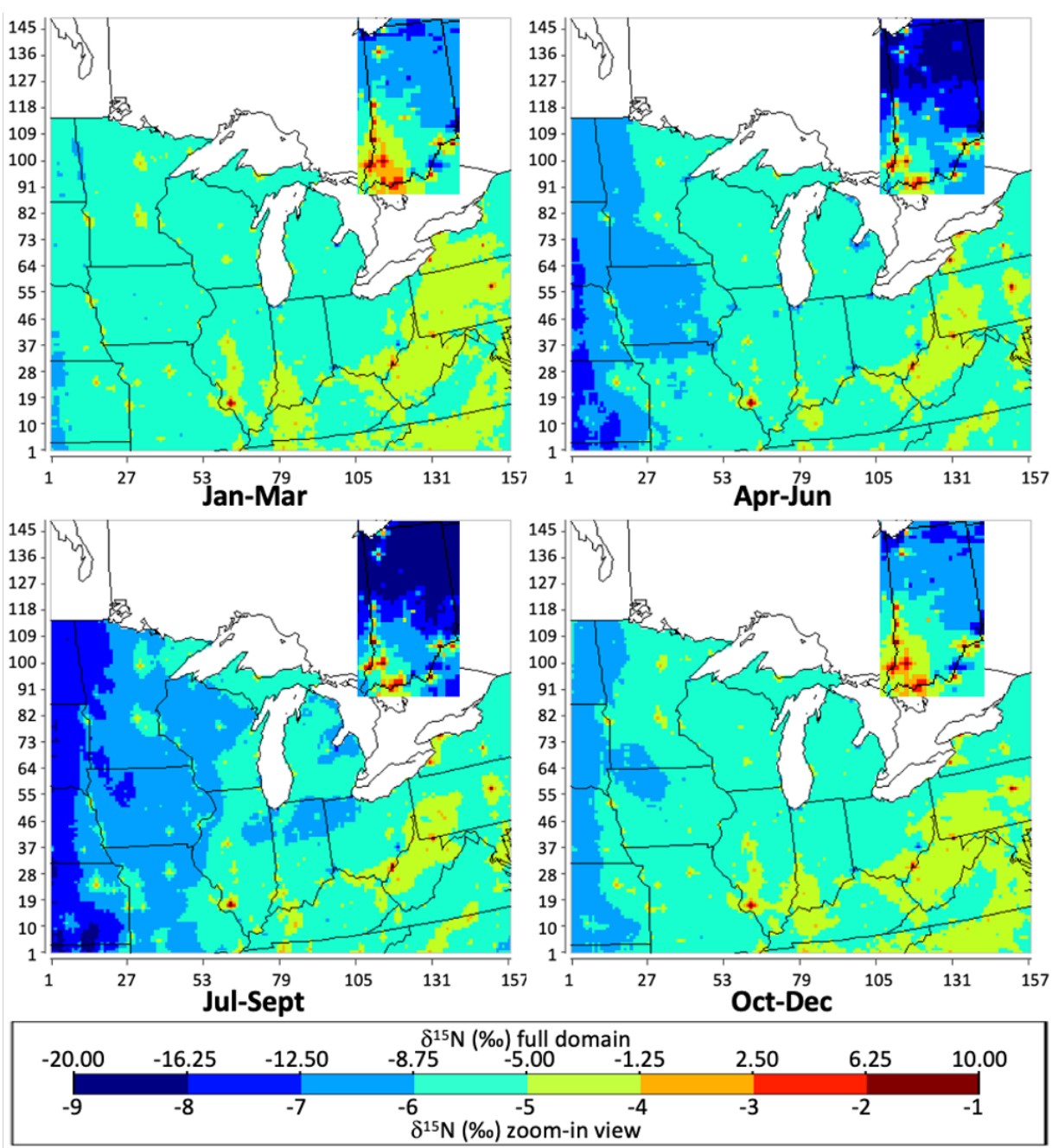

Figure 8: The geographical distribution of the $\delta^{15}N$ value of atmospheric $NO_x$ in each season (Winter: Jan-Mar; Spring: Apr-Jun; Summer: Jul-Sep; Fall: Oct-Dec) in per mil (‰) throughout the Midwest (with zoom-in view focusing on Indiana) simulated by CMAQ, based on NEI-2002 and 2016 meteorology.

Comparing the distributions of the difference in $\delta^{15}N$ values (Fig. S9) with the
corresponding PBL height (Fig. S10) among the maps of each season, the effects of PBL height
on the propagation of the air mass are clearly shown. The PBL height changes significantly
among each season within the geographic domain, especially over Minnesota, Wisconsin, and
Iowa (Fig. S10). The PBL height over these areas increases from less than 250 meters above the
ground level to more than 625 meters above the ground level, during spring (Apr-Jun) and
summer (Jul-Sep), which creates a more favorable synoptic condition for the dispersion, mixing,
and transport of the pollutants after being emitted into the atmosphere. As a result, the difference
in $\delta^{15}N$ values shifts to higher values, showing the stronger effect of atmospheric processes
during spring and summer. In order to qualitatively analyze how PBL height affects the level of
the dispersion, mixing, and transport of the pollutants, the average $\delta^{15}N$ value of atmospheric
$NO_x$ along the plumes of power plants was compared with the domain average PBL height for
each month within the Midwest region. The time series plot (Fig. 9a) shows the same seasonal
trend of $\delta^{15}N$ values along the power plants plumes and PBL heights over the domain.
Interestingly, the "turning point" of the $\delta^{15}N$ values is about one month later than the "turning
point" of the PBL heights. The scatter plot (Fig. 9b) shows a strong positive correlation between
the domain average PBL height and average $\delta^{15}N$ value along the power plants plumes, with
$R^2=0.85$. The positive correlation between PBL height and propagation of air mass, indicated by
the evolution of atmospheric $\delta^{15}NO_x$ in this study, agrees well with the corresponding
measurement in megacities in China from the previous studies (Shu et al., 2017; Liu et al., 2018;
Liao et al., 2018).

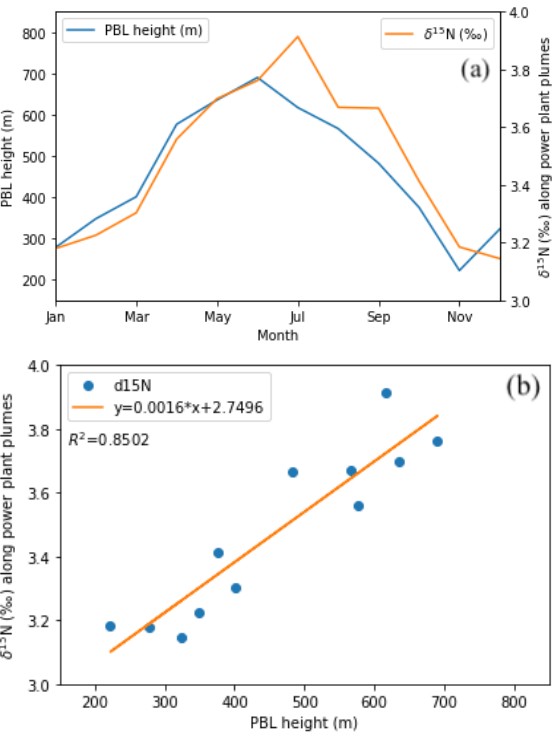

Figure 9: The time series plot (a) and the scatter plot (b) of the domain average PBL height (m) and the average $\delta^{15}N$ (‰) value of atmospheric $NO_x$ along the plumes of power plants during each month throughout the Midwest simulated by CMAQ, based on NEI-2002 and 2016 meteorology.

The atmospheric $\delta^{15}NO_x$ simulated based on different meteorology input dataset varies. In
order to compare the spatial heterogeneity of the atmospheric $\delta^{15}NO_x$ under different meteorology
conditions, the same analysis was done on the simulation using 2002 meteorology (Fig. S12).
Overall, the simulated atmospheric $NO_x$ under 2002 meteorology has the similar geographic
distribution and seasonal trend as the 2016 simulation. In order to qualitatively compare the
propagations of the pollutants under the impact of PBL height, the same plots were generated for
simulation based on 2002 meteorology (Fig. 10). Comparing to the two simulations (Fig. 10a)
reveals a similar seasonal trend but stronger monthly variation. Starting with lower PBL height
during the winter, the corresponding $\delta^{15}N$ values along the power plants' plumes were lower,
comparing to the simulation based on 2016 meteorology. As a result, the $\delta^{15}N$ values during the
spring and summer were also relatively lower. On the other hand, due to the higher PBL height
during the spring and summer for the simulation based on 2002 meteorology, the $\delta^{15}N$ values
decreased slower since July, ending with the relatively higher $\delta^{15}N$ values in December. The scatter
plot for the simulation based on 2002 meteorology (Fig. 10b) also shows a strong positive
correlation between the domain average PBL height and average $\delta^{15}N$ value along the power plants
plumes, with $R^2$=0.78. The videos of atmospheric $\delta^{15}NO_x$ on an hourly basis throughout the years
2002 and 2016 are available on Zenodo.org (10.5281/zenodo.4311986).

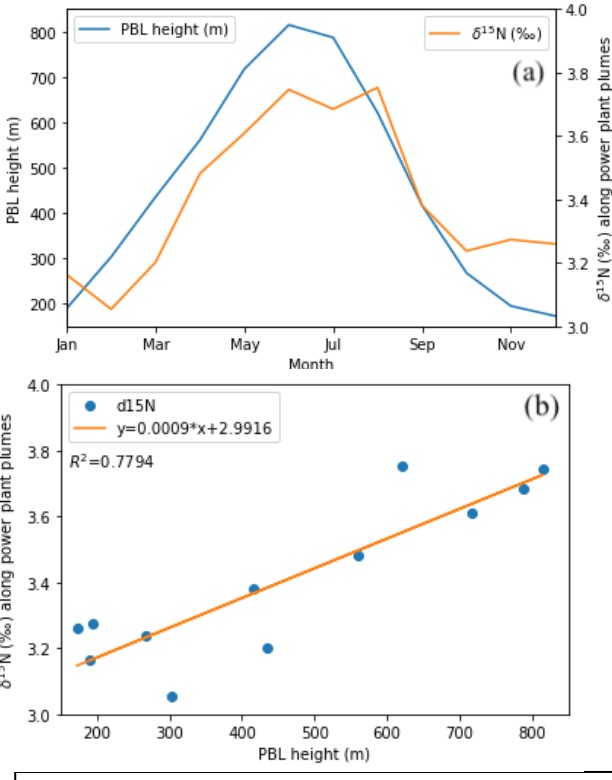

Figure 10: The time series plot (a) and the scatter plot (b) of the domain average PBL height (m) and the average $\delta^{15}N$ (‰) value of atmospheric $NO_x$ along the plumes of power plants during each month throughout the Midwest simulated by CMAQ, based on NEI-2002 and 2002 meteorology.

3.4 The simulation over the extracted domain
The temporal heterogeneity of difference in atmospheric $\delta^{15}NO_x$ between extracted-domain
simulation and full-domain simulation ($\Delta\delta^{15}N_{extracted\text{-}full}$), to explore the potential bias due to the
motion of the air mass across the boundary of the geographic domain of the study (Fig. 11). The
extracted domain covers the states of Indiana, Illinois, Ohio, and Kentucky, where the
measurements of $\delta^{15}NO_3^-$ at NADP sites are available. The predicted $\delta^{15}N$ of atmospheric $NO_x$
over the extracted domain shows a similar overall pattern as the $\delta^{15}N$ within the same domain from
the full-domain simulation, except over the southern border of the domain (Fig. S14). In order to
qualitatively analyze the effects from the initial boundary condition, the $\delta^{15}N$ of atmospheric $NO_x$
within IN, IL, OH, and KY were extracted from the full-domain simulation (Fig. 8) and compared
with the extracted-domain simulation within the same region (Fig. 11). The $\Delta\delta^{15}N_{extracted-full}$ values
ranged between -0.25‰ and +0.25‰ over most of the grids within the extracted domain, showing
the difference between extracted-domain simulation and full-domain simulation of $\delta^{15}N$ values are
trivial. However, near the southern border of the extracted domain, the obvious $\Delta\delta^{15}N_{extracted-full}$
values close to +0.75‰ during fall and winter, close to +1.00‰ during spring and summer occur,
which indicate the atmospheric $NO_x$ from the extracted-domain simulation is isotopically heavier.
The values of $\Delta\delta^{15}N_{extracted-full}$ become obvious near the southern border, which indicates the
dilution of $NO_x$, after it transports out of the domain since the $\delta^{15}N$ on the boundary was set to
zero. Unlike the southern border, the northern, western, and eastern border of the extracted domain
is located a sufficient distance apart from the boundary of the full domain. As a result, the
$\Delta\delta^{15}N_{extracted-full}$ values are similar over the majority grids within the domain.

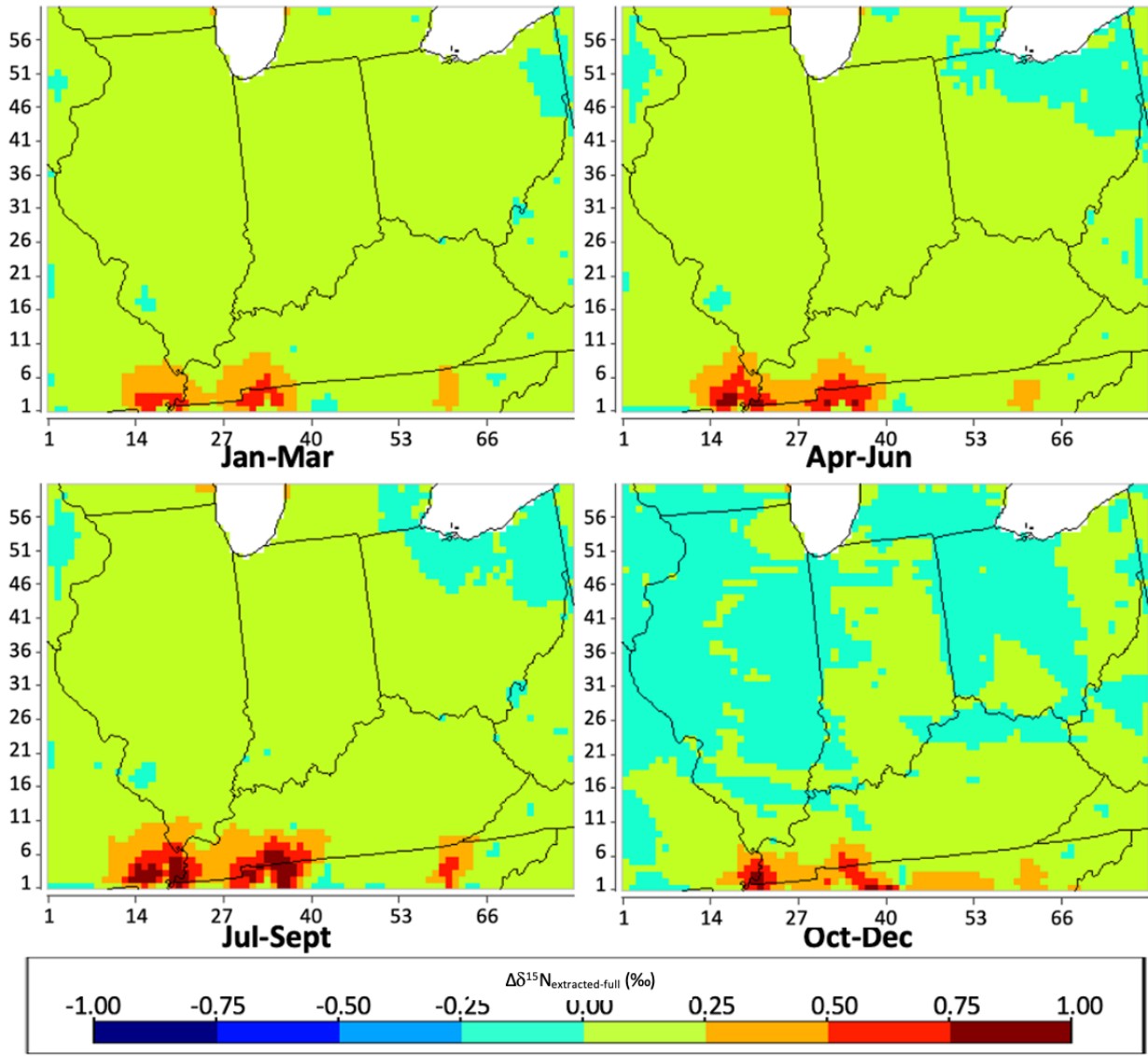

Figure 11: The geographical distribution of the difference between extracted-domain simulation and full-domain simulation of $\delta^{15}N$ value of atmospheric $NO_x$ ($\Delta\delta^{15}N_{extracted-full}$) in each season (Winter: Jan-Mar; Spring: Apr-Jun; Summer: Jul-Sep; Fall: Oct-Dec) in per mil (‰) within IN, IL, OH, and KY, based on NEI-2002 and 2016 meteorology.

2
3    3.5 The role of enhanced $NO_x$ deposition

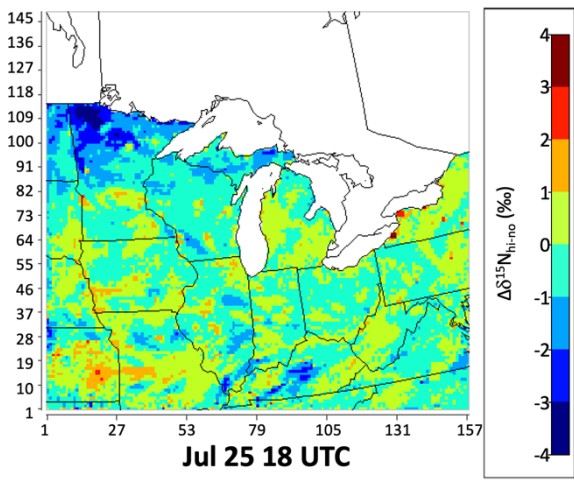

**Jul 25 18 UTC**

Figure 12. The $\Delta\delta^{15}N_{hi\text{-}no}$ values at 18 UTC on July 25.

The "emission + mixing + enhanced deposition" simulations significantly alter the $\delta^{15}N$ of atmospheric $NO_x$ relative to the normal deposition scenarios. Again, the enhanced deposition cases are removing $NO_x$ at rates that would be similar to those by removal during its conversion into $HNO_3$. Thus, in these cases the $NO_x$ deposited is $\sim \delta^{15}NO_3^-$ and the $\delta^{15}NO_x$ is that in the residual $NO_x$. The impact of high deposition on the residual $NO_x$ was assessed using $\Delta\delta^{15}N_{hi\text{-}no}$ , the difference between the $\delta^{15}NO_x$ values of atmospheric under the "enhanced deposition" and "no deposition" scenarios. The $\Delta\delta^{15}N_{hi\text{-}no}$ range was $\pm 4‰$ and was especially obvious downwind of the locations with large emission rates, such as power plants or megacities (Fig. 12a).  This can be explained as a similar fashion to the "no deposition" scenarios (Fig. S15a), where the dispersion of the isotopically heavier $NO_x$ emission from big cities, major highways, and power plants elevates the $\delta^{15}NO_x$ values in the surrounding grids located in rural areas, the dispersion of the isotopically lighter biogenic $NO_x$ emission lowers the $\delta^{15}NO_x$ values in the surrounding grids located in the suburb of major cities (Fig. S15b). On the other hand, due to the higher deposition rate, the transport, mixing, and dispersion of $NO_x$ emission from different sources are restricted within a smaller geographical extent (Fig. S15b). As a result, under the "enhanced deposition" scenario, the $NO_x$ emissions disperse to fewer surrounding grids but lead to a lower $\delta^{15}NO_x$ values relative to no deposition. The temporal heterogeneity of $\Delta\delta^{15}N_{hi\text{-}no}$ over the domain was examined and the impact of enhancing deposition rates of $NO_x$ on the $\delta^{15}N$ of atmospheric $NO_x$ on the seasonal basis was explored (Fig. 14). The seasonal $\Delta\delta^{15}N_{hi\text{-}no}$ values range from -3.67‰ to 5.34‰, with an average of 0.51‰. The overall pattern of the $\Delta\delta^{15}N_{hi\text{-}no}$ values shows that due to deposition, the atmospheric $NO_x$ became isotopically lighter over the majority of the grids since EGU and vehicle $NO_x$ is not being transported as far in the enhanced deposition. Conversely, in grids that contain or surround power plants and big cities the $\delta^{15}NO_x$ increases because it is not as effectively mixing with low $\delta^{15}NO_x$ from nearby grids. The enhanced deposition simulation somehow presents the isotope effects associated with the "pseudo photochemical transformation" of $NO_x$ into $NO_y$. The complete isotope effect of tropospheric photochemistry will be addressed in future work, which incorporates $^{15}N$ into the chemical mechanism of CMAQ for the simulation.

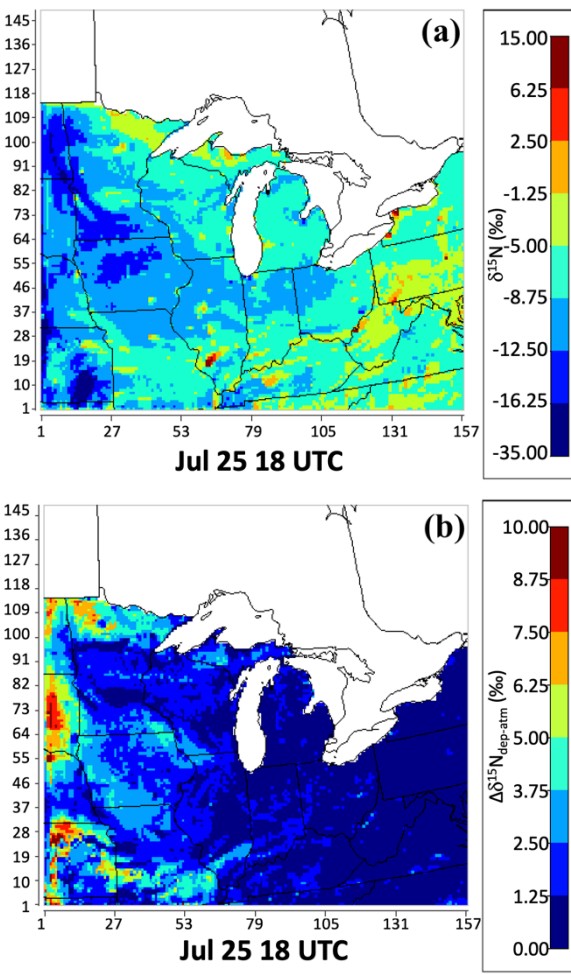

Figure 13: The $\delta^{15}N$ values of $NO_x$ deposition under the "enhanced deposition" scenario (a); the $\Delta\delta^{15}N_{dep-atm}$ (b), at 18 UTC on July 25, are presented by color in each grid (NEI-2002 and 2016 meteorology). The warmer the color, the higher $\delta^{15}N$ and $\Delta\delta^{15}N_{deposition}$ values of atmospheric $NO_x$

The $\delta^{15}NO_x$ deposition (proxy for $\delta^{15}NO_3^-$) simulated by CMAQ at these sites show similar
monthly variations and seasonal trends as SMOKE (Fig. S19). The ranges of $\delta^{15}N(NO_x)$ values
within each month were narrower, comparing to the simulation from SMOKE, with a minimum
during February (-8.7~ -4.4‰) and a maximum during August (-11.8~-4.2‰). The seasonal trend
shows low $\delta^{15}N(NO_x)$ during summer, with the median around -7.4‰, and high $\delta^{15}N(NO_x)$ during
winter, with the median around -6.0‰. Therefore, the CMAQ simulation inherits the monthly
variations and seasonal trends from SMOKE, while the atmospheric $NO_x$ becomes isotopically
heavier, after taking atmospheric mixing and transport into account. As mentioned above, most of
the NADP sites are located away from big cities and power plants. Thus, the atmospheric mixing
and transport led to the isotopically heavier atmospheric $NO_x$.

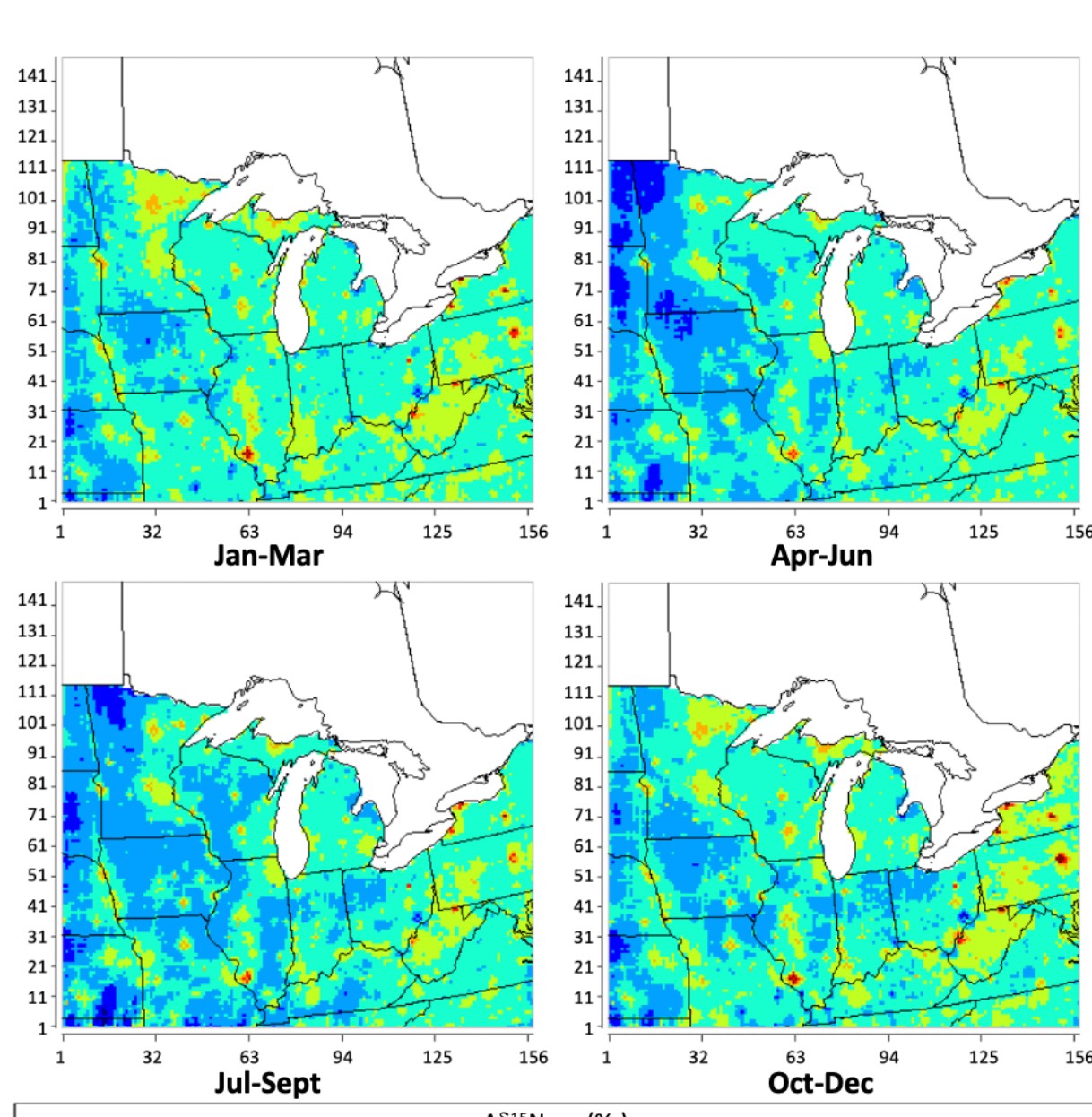

**Jan-Mar**

**Apr-Jun**

**Jul-Sept**

**Oct-Dec**

$\Delta\delta^{15}N_{hi-no}$ (‰)

-4    -3    -2    -1    0    1    2    3    4

Figure 14: The difference between the $\delta^{15}N$ (‰) value of atmospheric $NO_x$ under the "enhanced deposition" scenario and "no deposition" scenario ($\Delta\delta^{15}N_{hi-no}$) during each season (Winter: Jan-Mar; Spring: Apr-Jun; Summer: Jul-Sep; Fall: Oct-Dec), throughout the Midwest simulated by CMAQ, based on NEI-2002 and 2016 meteorology.

3    3.6    Model-observation comparison of $\delta^{15}NO_x$

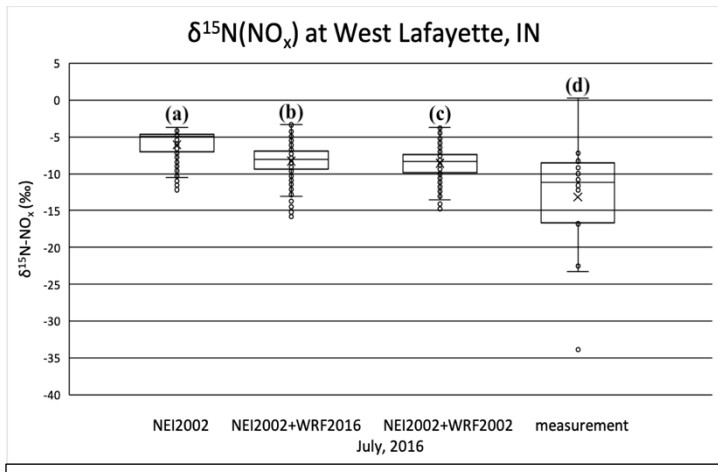

Figure 15: The $\delta^{15}NO_x$ distributions for Lafayette, IN from July 8 to August 5, simulated by SMOKE (a), CMAQ based on 2016 (b) and 2002 meteorology (c), compare with the measured $\delta^{15}NO_x$ (d) taken on July to August in 2016 (box: lower quartile, median, upper quartile; whisker: lower extreme, upper extreme; dots outside the whisker: outliers)

In order to evaluate the SMOKE/CMAQ simulations of atmospheric $\delta^{15}N$, they were compared to several existing observational datasets. The $\delta^{15}N$ values under the "no transport" simulation by SMOKE in West Lafayette, IN was compared with the measurement (Walters, Fang, & Michalski, 2018) from July 8 to August 5, 2016 (Fig. 15). The range of SMOKE simulated $\delta^{15}N(NO_x)$ from NEI-2002 ranges from -12.2‰ to -3.8‰, which is within the range of the corresponding measurement (-33.8 ~ 0.2 ‰). Whereas the median (-5.0 ± 2.2 ‰) of SMOKE simulated $\delta^{15}N(NO_x)$ is higher than the median (-11.2 ± 8.0 ‰) of the measured values. The SMOKE simulated $\delta^{15}N(NO_x)$ values in West Lafayette, IN are higher than the corresponding measurements. Therefore, the emission from the soil, livestock waste, off-road vehicles, and natural gas power plant might be underestimated, and/or the emission from the on-road vehicle and coal-fired power plant might be overestimated for both versions of NEI.

In addition to the effects from $NO_x$ emission sources, the lower values and greater variations in measured $\delta^{15}N(NO_x)$ might also be caused by the atmospheric mixing with the emission from surrounding grids, driven by the atmospheric processes. The $\delta^{15}N$ of atmospheric $NO_x$ under the "with transport" scenario by CMAQ with different meteorology conditions (simulated by WRF for the year 2002 and 2016) was compared with the measurement (Walters, Fang, & Michalski, 2018) from July 8 to August 5, 2016 (Fig. 15). The $\delta^{15}N$ of atmospheric $NO_x$ simulated based on 2016 meteorology ranges from -15.8‰ to -3.4‰, with the medium of -8.1 ± 2.1‰; the $\delta^{15}N$ of atmospheric $NO_x$ simulated based 2002 meteorology ranges from -14.8‰ to -3.7‰, with the medium of -8.4 ± 1.9‰. The $\delta^{15}N$ of the corresponding measurement ranges from -33.8‰ to 0.2‰, with the medium of -11.2 ± 8.0‰. In general, the CMAQ simulations of $\delta^{15}N(NO_x)$ under both of the scenarios conducted in this study perform better than the SMOKE simulation of $\delta^{15}N(NO_x)$, which only takes the variability of the $NO_x$ emission source into account (Table S7).

3.7 Enhanced $NO_x$ deposition simulating $\delta^{15}NO_3$: model observation comparison

The model was used to predict $\delta^{15}NO_3^-$ and compared with the $\delta^{15}NO_3^-$ in deposition collect between 2001 and 2003 at several Midwestern NADP sites (Table S4). The measurements of $\delta^{15}N$ values of $NO_3^-$ at NADP sites from prior studies (Mase, 2010; Riha, 2013) show the similar monthly variations and seasonal trend as both "no transport" and "with transport" simulations (Fig. S19). There is a wide range of $\delta^{15}N(NO_3^-)$ values within each month, with a minimum during January (10.4~17.2‰) and a maximum during August (1.0~16.7‰). The seasonal trend shows low $\delta^{15}N(NO_3^-)$ during spring, with the median around 9.3‰, and high $\delta^{15}N(NO_3^-)$ during winter, with the median around 13.0‰. The measured $\delta^{15}N$ values of $NO_3^-$ have the same seasonal trend as the simulated $\delta^{15}N$ values of $NO_x$. Even considering the effect of atmospheric mixing and transport, the measured $\delta^{15}N$ values of $NO_3^-$ is about 17‰ higher than the simulated $\delta^{15}N$ values of $NO_x$. The difference between CMAQ simulated and measured $\delta^{15}N$ values of deposition is caused by the following two factors: a). the mixture of isotopically lighter $NO_x$ from the surrounding area discussed in section 3.3, and b). the net N isotope effect during the conversion of $NO_x$ to $NO_3^-$, which will be addressed in future work.

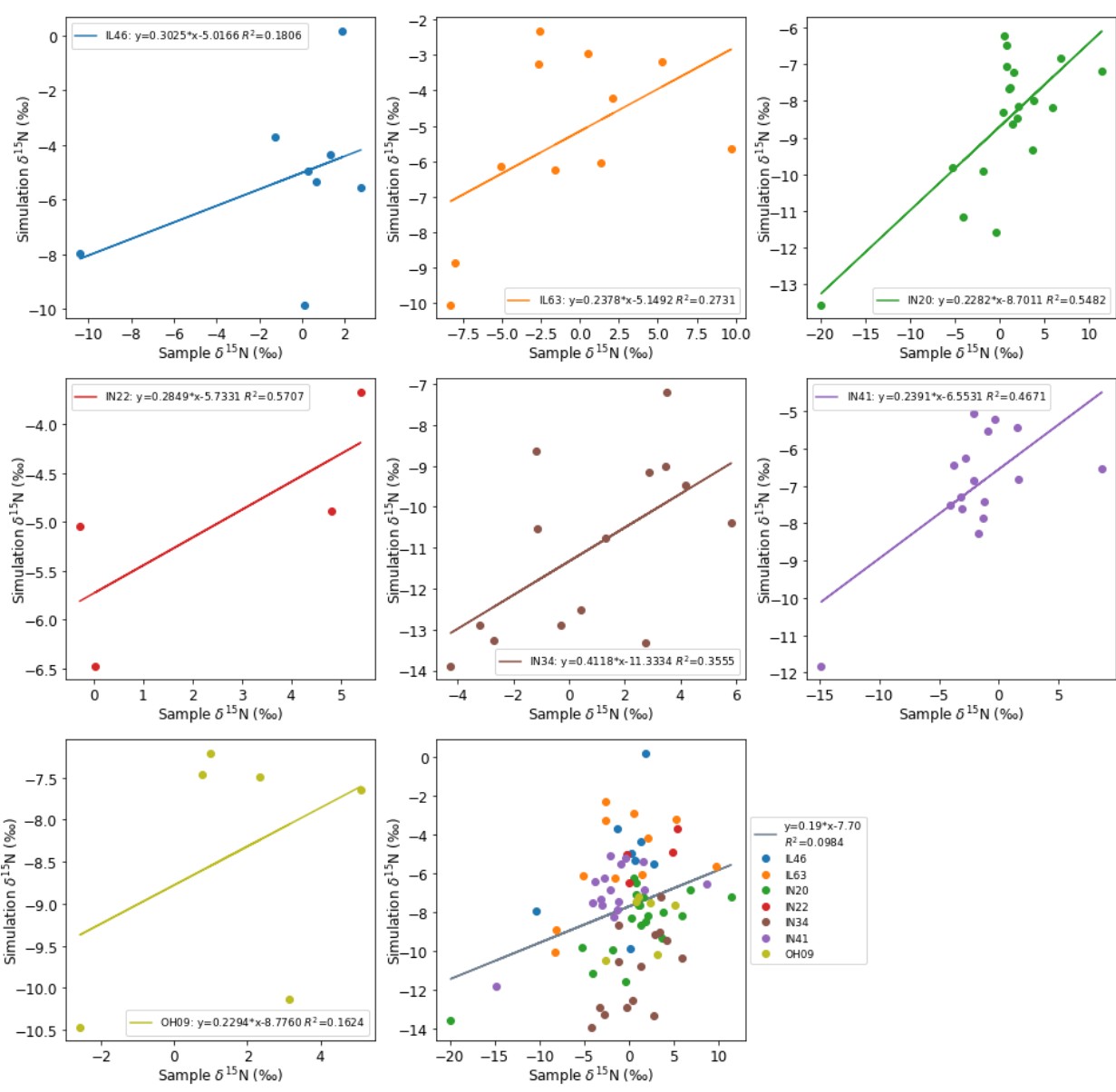

Figure 16: The emission + mixing + deposition CMAQ predicted $\delta^{15}N$ value of $NO_x$ deposition using NEI-2002 and 2002 meteorology compared to the measured $\delta^{15}N$ of rain $NO_3^-$ at NADP sites within IN, IL, and OH.

1        The 30 fold enhanced $NO_x$ deposition (see methods) was used to simulate the $\delta^{15}N$ value of
$NO_3^-$ deposition ($\delta^{15}N(NO_3^-)$) that was then compared to observations (Fig. 16). As previously
noted, rather than explicitly converting $NO_x$ into $NO_y$ via the chemical mechanism in CMAQ,
which would require writing an isotope-enabled chemical scheme with appropriate rate constants,
we amplified $NO_x$ deposition as a surrogate. This amplification reduced the $NO_x$ lifetime to about
1 day, thus by calculating the $\delta^{15}N$ of $NO_x$ in the deposition fraction, as opposed to residual $NO_x$
in the atmosphere, we are approximating the $\delta^{15}N(NO_3^-)$ in deposition. The model approximation
was compared to $NO_3^-$ collected at NADP sites within Indiana, Illinois, and Ohio in the year 2002
(Table S4). The NEI-2002 and WRF2002 were used for the SMOKE emission model and CMAQ
simulations, respectively. The $\delta^{15}N(NO_3^-)$ value in deposition was calculated by $\delta^{15}N(NO_3^-) = \Sigma$
$f_{NOxhr}$ $\delta^{15}N(NO_x)_{hr}$, where $f_{NOxhr}$ is the hourly mole fraction of $NO_x$ isotopologue deposited ($f_{NOxhr}$
= $NO_{xhr}/NO_{xT}$) and $\delta^{15}N(NO_x)_{hr}$ is the $\delta^{15}N$ value of $NO_x$ in deposition. The total $NO_x$ deposited
($NO_{xT}$) used to calculate $f_{NOxhr}$ was the amount deposited 5 days prior to the sampling date since
the NADP deposition collection integrate the week. The $\delta^{15}N$ values of $NO_x$ deposition simulated
by CMAQ under the "enhanced deposition" scenario at each site were compared with the
measurements of $\delta^{15}N$ values of $NO_3$ from prior studies (Mase, 2010; Riha, 2013). The scatter
plots show moderate positive correlation between sample $\delta^{15}N$ and simulated $\delta^{15}N$, with $R^2$
between 0.16 and 0.57 (Fig. 16). The difference in the trend line equations among the NADP sites
might be caused by the difference in air temperature and photolysis rate, which impact the chemical
mechanisms converting $NO_x$ into $NO_y$ and will be explored in future study.
4.   Conclusion
13       The $\delta^{15}N$ of atmospheric $NO_x$ was simulated by SMOKE, by considering the $NO_x$ emissions
from NEI emission sectors and the corresponding $\delta^{15}N$ values from previous research. $\delta^{15}N$ is an
effective tool to track the atmospheric $NO_x$, in terms of its evolution of spatial and temporal
composition, altered by atmospheric processes The simulation indicates that the $NO_x$ emission
from biogenic sources is the key driver for the variation of $\delta^{15}N$, especially among the NADP sites.
The uncertainties in the $\delta^{15}N(NO_x)$ simulation are less than 5‰ over the majority of the grids
within the Midwest. For the $NO_x$ emission from the regions dominated by biogenic source, the
uncertainties in the $\delta^{15}N(NO_x)$ simulation are less than 10‰. The uncertainties in the $\delta^{15}N(NO_x)$
simulation were well below the difference among the $\delta^{15}N(NO_x)$ values from different $NO_x$
emission sources (Fig. S20). Comparing with the measurements of $\delta^{15}N(NO_3^-)$ from NADP sites
within Indiana, Illinois, Ohio, and Kentucky, the simulated $\delta^{15}N$ agreed well with the seasonal
trend and monthly variation. While the simulated $NO_x$ is slightly heavier than the corresponding
measurements in West Lafayette, IN, taken from July to August 2016. According to the previous
research, the uncertainty of $NO_x$ emission is 71-250% from soil and 10-15% from the vehicle. The
variations among the removal efficiency of different emission control technologies vary from 30%
to 90%, also causes the uncertainty of power plant $NO_x$ emission. In addition, in this study, due to
the lack of measurements, the $\delta^{15}N$ of coal-fired and natural gas non-EGUs (industrial boilers,
commercial and residential fuel combustions) were assumed to be the same as the $\delta^{15}N$ of coal-
fired and natural gas EGUs respectively. Thus, detailed measurements of the $\delta^{15}N$ of non-EGUs
are necessary for future study. Besides this, the non-road vehicles (aircraft, ships, and trains) also
need to be included in the future study.
34       If we only consider the effects from $NO_x$ emission sources, the emission from soil, livestock
waste, off-road vehicles, and natural gas power plant in West Lafayette, IN are possible to be
underestimated, and the emission from the on-road vehicle and coal-fired power plant in West
Lafayette, IN are possible to be overestimated. Another reason causing the estimated $NO_x$
isotopically heavier than measured $NO_x$ is the mixing caused by atmospheric processes, since the
$NO_x$ emission from the surrounding region of West Lafayette, IN is lighter. In addition, the
tropospheric photochemistry could also alter the $\delta^{15}N$ values during the processes that convert $NO_x$
to $NO_y$.
42       After considering the impacts of atmospheric processes, by simulating CMAQ based on the
$^{15}N$ incorporated emission input datasets and the meteorology input dataset simulated from WRF
and MCIP, the performance of the simulated $\delta^{15}N(NO_x)$ is better. The simulation indicates that the
PBL height is the key driver for the mixture of anthropogenic and natural $NO_x$ emission, which
deepens the gap between $\delta^{15}N$ of atmospheric $NO_x$ and $NO_x$ emission. After considering the effects

of NO$_x$ emission sources and atmospheric processes, there is still an obvious gap between the simulated $\delta^{15}N(NO_x)$ and the corresponding measurements. Therefore, before adjusting the NO$_x$ emission inventory, the future work is to explore how tropospheric photochemistry alters $\delta^{15}N(NO_x)$ by incorporating $^{15}N$ into the chemical mechanism of CMAQ and comparing the simulation with the corresponding measurements. With the validation of our nitrogen isotopes incorporated CMAQ, the NO$_x$ emission inventories could be effectively evaluated and improved.

**Data availability:** The source code for SMOKE version 4.6 is available at https://github.com/CEMPD/SMOKE/releases/tag/SMOKEv46_Sep2018. The source code for CMAQ version 5.2.1 is available at https://github.com/USEPA/CMAQ/tree/5.2.1. The in-detail simulation results for $\delta^{15}N$ of NO$_x$ emission based on 2002 and 2016 versions of National Emission Inventory and the associated python codes are achieved on Zenodo.org (10.5281/zenodo.4048992). The input datasets for WRF simulation are available at https://www.ncei.noaa.gov/data/. The in-detail simulation results for $\delta^{15}N$ of atmospheric NO$_x$ under all scenarios discussed in this paper and the CMAQ-based c-shell script for generating BCON for extracted domain simulation are achieved on Zenodo.org (10.5281/zenodo.4311986).

**Author contributions:** Huan Fang and Greg Michalski were the investigator for the project and organized the tasks. Huan Fang develop the model codes, performed the simulation to incorporate $^{15}N$ into SMOKE outputs and generated $\delta^{15}N$ values and reconstruct CMAQ by incorporating $^{15}N$, and performed the simulation to generate $\delta^{15}N$ values. Greg Michalski helped Huan Fang in interpreting the results. Huan Fang prepared the manuscript with contributions from all co-authors.

**Acknowledgments:** We would like to thank the Purdue Research Foundation and the Purdue Climate Change Research Center for providing funding for the project. We would like to thank Scott Spak from School of Urban & Regional Planning, University of Iowa for simulating SMOKE using NEI-2002. We would like to thank Tomas Ratkus from Department of Earth, Atmospheric, and Planetary Sciences, Steven Plite, and Frank Bakhit from Rosen Center for Advanced Computing, Purdue University for setting up CMAQ on Purdue research computing for this project.

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
