# Peer review of "Assessing the roles emission sources and atmospheric processes play in simulating δ15N of atmospheric NOx and NO3 using CMAQ (version 5.2.1) and SMOKE (version 4.6). Huan Fang† and Greg Michalski†‡ 6 †Department of Earth, Atmospheric,"

_Geoscientific Model Development, 2020_

## Referee Comment (RC2)

Fang and Michalski have put forth a lot of hard work paving the way for $\delta^{15}N$ values to be modeled correctly for regional-scale evaluation of $NO_x$ emissions, chemical mechanisms, atmospheric transport, and deposition rates. Based on the importance of the topic of using stable isotopes to probe our understanding of atmospheric processes and the general soundness of their approach, I am in favor of this study eventually being published in GMD. However, I think major changes are necessary before it should be accepted as detailed below.

General Comments:

1. I agree with the first reviewer about the scope of this paper being too narrow, and I think that reviewer's suggestion of combining this work with Fang and Michalski (2020) would be appropriate. An alternative solution would be to keep these papers separate but extend this paper to consider chemistry and aerosol processes in addition to transport and deposition. This alternative solution would allow the authors to keep the focus of each paper narrowed to the model application that is being used (i.e. SMOKE and CMAQ respectively) rather than mixing the two. There are a lot of details and useful figures presented in Fang and Michalski (2020) that warrant its own paper in my view. I suspect that the chemistry mechanism is still under development and/or characterization or it would have been included here. Unfortunately, it is hard for me to view this current paper as a complete study without that critical piece. If it can be added, I think the results shown here for the transport-only cases are still somewhat useful to present, especially in a journal like GMD. As presented though, I think the paper slices the development steps too thin to be as impactful as the authors would hope.

2. I have serious concerns about the overlap in content with Fang and Michalski (2020). Fig. 1 must be removed and the introduction before Page 3, line 14 should be rewritten to focus on issues of transport and deposition (and chemistry) if that is to be the focus of the methods and results. The reader should be referred to Fang and Michalski (2020) for a discussion of sources. Feel free to reprint the data in Fig. 1 as a table in the SI if you want it to travel with the paper, but it's inappropriate to use it again in the main text.

3. I am not sure what we are learning from the many figures showing the seasonal variation in $\delta^{15}N$ concentrations for transport only, transport with different emissions, transport with different meteorology, and transport with deposition on, etc. For example, Fig. 9 could probably be one map since there isn't much variability among seasons. The conclusion that the PBL is the "key driver for the mixture of anthropogenic and natural $NO_x$ emission" seems odd. Are the authors highlighting PBL height to distinguish vertical dispersion from horizontal advection and deposition? If so, I would think a more systematic approach would be to show box-whisker plots for all the $\delta^{15}N$ values like in Fig. 13 but with only vertical mixing and emissions on, then with horizontal advection and dispersion (and vertical advection) added, and finally with deposition on and show which is having the largest overall impact. The authors could also add the cases with varied emissions and meteorology, but most importantly, all of these sensitivities need to be summarized visually somehow for the message to come through.

4. It seems like the measurements could be leveraged to evaluate the model more directly (see specific comments below). For example, is the model getting the day-night trend correct at West Lafayette?

5. I second the first reviewer about the grammar issues throughout the text. I am happy to provide specific comments on a future draft after the major issues are resolved or justified.

Specific Comments:

1. Please add quantitative metrics to the abstract to more precisely communicate the impact that adding CMAQ's process-level understanding has on the evaluations in Indiana.

2. Page 2, Line 31: Better add the reference to the FIVE mobile emission model (McDonald et al., 2018: https://doi.org/10.1021/acs.est.8b00778) to these references. It would also be good to add a more recent reference for the MOVES model from US EPA.

3. Page 3, Line 18: Consider changing "$NO_x/NO_y$" to "$NO_x$" or "$NO_x$ and $NO_y$"

4. Section 2.1: Why run with an extracted domain? Was this just to make the model go faster?

5. Consider moving most of section 2.2 to SI since it is covered in the companion paper.

6. Section 2.3: Please specify model version numbers for WRF, SMOKE, and MCIP in this section. Much of section 2.3 can be moved to SI. The main manuscript can just state what models and version numbers were used for each part. The details of how the data were processed can be included in the SI, especially because I understand they probably took an enormous amount of the authors' time during this project. But as presented, I think they dilute the narrative of isotopic $NO_x$ that the authors want to stick to.

7. Page 10, Lines 3-18: It is hard to understand exactly what was accomplished in the deposition velocity tuning approach and what its limitations are. This would all be solved if the authors were able to include chemistry in the study and turn chemistry off for a transport+deposition only case.

8. Section 2.6: We need some idea of how the emission datasets performed against coincident observations from routine networks for conventional pollutants like NO2, EC, O3 and particulate Sulfate to check that they were processed with reasonable assumptions.

9. Please consider removing Tables S1 and S2 from the supplemental information. Just refer readers to the MCIP user guide.

10. Section 2.7: I'm not sure what is meant by the 'research area' and 'emission-free zone'. Is it just U.S. versus Canada? The term 'nested' usually refers to an area of higher resolution. Although this doesn't strictly have to be the case, I urge the authors to consider renaming their 'nested' grid to the 'research area' or something similar, to identify that this is the area they are using for their analysis as it is far from the interfering model boundaries.

11. Page 14: Rather than using Fig. 3 to show the expected dispersion of $NO_x$ in the model domain, why not include a figure as a barplot that quantifies the differences in weighted average $\delta^{15}N$ values for the different categories discussed like agricultural areas, big cities, highways and EGUs?

12. Page 14, line 28: Consider adding a figure with distance from power plant on the x and $\delta^{15}N$ on the y to show the decay along a couple of trajectories from an important facility.

13. Page 18: Is there any data to evaluate the PBL heights? This is a critical part of the study the way the authors have framed it. Also which PBL model scheme in WRF and CMAQ did the authors use? Maybe they should try a different one?

14. Page 18, line 12: Why not have a figure showing the positive correlation between PBL height and $\delta^{15}N$ or whatever would correspond to the Chinese studies to demonstrate the consistency check.

15. Fig 7 can go to the SI.

16. Fig. 10 is not terribly informative other than to show that the southern boundary should probably also be restricted for the nested area. The authors should consider redefining their nested area with this in mind for all future simulations on this domain.

17. I recommend moving Section 3.7 to the first section of the Results. I find it helpful to start with the model-obs comparison and then dive into model predictions that are unconstrained by observations.

18. Strongly recommend putting Figs. 11 and 12 together so that measurements and models are on the same figure. Why not pair the model and obs in time for the figure? Page 28 text should be moved to methods section. Figure 12 and associated text does not really belong in this section as it is not a model-obs comparison.

19. Fig. 14: Consider normalizing both panels to remove influence of chemistry bias. It seems like there is a signal here that the CMAQ modeling is able to capture, but it is hard to tell.

---

## Author Response (AR1)

**RC1**

All typo and grammatical errors were adjusted according to the comments, the suggested citations were added. The responses to other comments are included below.

1. "… the authors themselves point out on page 4 (lines 11-15) the "unsatisfactory" nature of the 2020 paper's approach so why is it justified to separately publish that?", "It is of critical importance to address the significant overlap between this work and the cited companion paper of Fang & Michalski (2020) …"

We decided to combine the companion paper with this manuscript in order to deal with the overlapping and completeness issue.

2. "… While I value the need to focus on the detailing the model specifics in a journal such as GMD, the manuscript here lacks any real interpretation, quantification of the sensitivity of the output to the model parameters and consideration of the implications of the predicted values compared to previous studies in the literature of interpreting the isotopic composition of NOx and nitrate …"

The more in detail interpretation and quantification of the output have been included in the revised manuscript (In section 3.2, 3.3, 3.5, and 3.7)

3. "Second, the model is compared with one set of observations of "d15N-NOx" in Indiana (within the domain of the model runs) …"

The full simulation domain covers the whole Midwest while the sampling sites only locate in IN, IL, OH, and KY. As a result, the sampling site would have sufficient distance between the domain boundary to eliminate the bias near the domain boundary.

4. "Additionally, the measurements that are compared with are specifically d15N-NO2 and not d15N-NOx …"

Walters et al (2018) did include the d15N(NOx) values based on the measurements in Table 1 and 3.

"The first comparison is to the only direct measurements within the domain, which occurred in West Lafayette, IN. The $\delta15N(NOx)$ values were inferred from the measured $\delta15N(NO2)$ and the calculated $\delta15N(NO2)$ shift (Walters, Fang, & Michalski, 2018)."

5. "Third, the sensitivity to the starting emissions values should be evaluated …"

The range of d15N values for any source is generally a function of equilibrium, kinetics, or reaction progress happening in that source process. For example, automobiles show a wide range of both NOx amount and d15N values going from cold start to normal driving, but once the catalytic converter is warm the values are relatively constant because the NOx reduction by the CC becomes constant. We are using the average to account for these effects and for simplicity.

In future work hope to explicating model the sources variation in SMOKE or land surface models, but that is well beyond the scope of this work. Fig. S17 shows the uncertainties of d15N values within the research area. For most of the grids, the uncertainties are less than 5 ‰, which is well below the difference in d15N values between any two of the emission sources. For those regions dominated by biogenic source, the uncertainties in d15N values are less than 10 ‰, which is also significantly below the difference in d15N values between the emission from biogenic source and all the other sources.

6. "Fourth, one of the key conclusions of this work is that changes in the polluted boundary layer (PBL) are critical to transport and dispersion of NOx such that the pattern of d15N- NOx is importantly changed based on the PBL height. I'm not convinced the results shown support this conclusion …"

The more in detail interpretation and quantification of the relationship between d15N and PBL height has been included in section 3.3 of the revised manuscript

7. "Fifth, "the role of deposition" section and comparison of d15N-NOx with d15N-NO3- seems out of place in this work …"

CMAQ simulated the d15NOx effect by NOx removal using enhanced deposition. These "emission + mixing + enhanced deposition" simulations were not imposing an isotope effect related to dry/wet deposition, rather they are an attempt to show how "lifetime chemistry" alters NOx d15N values by removing NOx before it can be transported significant distances.

8. "Finally, it needs to be addressed why in this work there are only 8 NADP sites being compared with, while it appears that 82 measurement sites are included in FM2020?"

We decided to use the exact measurement from our lab at the 8 NADP site to validate the simulation, instead of using the values of literature review and compared with the simulation values at the grids contains the NADP site within the simulation domain.

9. "Title: Is it necessary to have the CMAQ, SMOKE and WRF versions as part of the title?"

The editor requested us to include the model's name and the version number in the title

10. "I would argue that atmospheric "processes" are not really being tested here, it's really transport or meteorology …"

Transport, mixing, dispersion, and deposition are all examples of atmospheric processes

RC2

1. "I agree with the first reviewer about the scope of this paper being too narrow, and I think that reviewer's suggestion of combining this work with Fang and Michalski (2020) would be appropriate ...", "I have serious concerns about the overlap in content with Fang and Michalski (2020). "
We decided to combine the companion paper with this manuscript in order to deal with the overlapping and completeness issue.

2. "I am not sure what we are learning from the many figures showing the seasonal variation in δ15N concentrations for transport only, transport with different emissions, transport with different meteorology, and transport with deposition on, etc ..."
The more organized interpretation and quantification of the output have been included in the revised manuscript in section 3

3. "The conclusion that the PBL is the "key driver for the mixture of anthropogenic and natural NOx emission" seems odd ..."
The more in detail interpretation and quantification of the output have been included in the revised manuscript in section 3.3

4. "Please add quantitative metrics to the abstract to more precisely communicate the impact that adding CMAQ's process-level understanding has on the evaluations in Indiana."
Confirmed

5. "Page 2, Line 31: Better add the reference to the FIVE mobile emission model (McDonald et al., 2018: https://doi.org/10.1021/acs.est.8b00778) to these references."
Confirmed

6. Page 3, Line 18: Consider changing "NOx/NOy" to "NOx" or "NOx and NOy"
Confirmed

7. "Section 2.1: Why run with an extracted domain? Was this just to make the model go faster?"
To eliminate the bias near the domain boundary

8. "Much of section 2.3 can be moved to SI. The main manuscript can just state what models and version numbers were used for each part."
Confirmed

9. "Page 10, Lines 3-18: It is hard to understand exactly what was accomplished in the deposition velocity tuning approach and what its limitations are. This would all be solved if the authors were able to include chemistry in the study and turn chemistry off for a transport+deposition only case."
Including chemistry in the study and turn chemistry off is what exactly we did. The more in detail explanation of deposition have been included in the revised manuscript in section 3.7.

10. "Section 2.6: We need some idea of how the emission datasets performed against coincident observations from routine networks for conventional pollutants like NO2, EC, O3 and particulate Sulfate to check that they were processed with reasonable assumptions."
We decide to give up this section, and explore the sensitivity of emission dataset in future work

11. "Please consider removing Tables S1 and S2 from the supplemental information. Just refer readers to the MCIP user guide."
Table S1 and S2 were summarized from user's guide, it is easier to understand for the reader and more relevant to the corresponding section in the manuscript.

12. "Section 2.7: I'm not sure what is meant by the 'research area' and 'emission-free zone'. Is it just U.S. versus Canada? The term 'nested' usually refers to an area of higher resolution. Although this doesn't strictly have to be the case, I urge the authors to consider renaming their 'nested' grid..."
We do not have emission dataset in Canada, thus when the air mass transports out of the Midwest, the atmospheric NOx is diluted. In addition, we set atmospheric NOx $\delta15N = 0‰$ for initial condition and boundary condition, the air mass from Canada, or boundary of the domain could flattened the atmospheric $\delta15N(NOx)$. Details in section 2.2.2-2.2.4. The term 'nested' was changed to 'extracted'

13. "Page 14: Rather than using Fig. 3 to show the expected dispersion of NOx in the model domain, why not include a figure as a barplot that quantifies the differences in weighted average $\delta15N$ values for the different categories discussed like agricultural areas, big cities, highways and EGUs?"
Fig. S6 in the revised manuscript

14. "Page 14, line 28: Consider adding a figure with distance from power plant on the x and $\delta15N$ on the y to show the decay along a couple of trajectories from an important facility."
Fig. 6 in the revised manuscript

15. "Page 18: Is there any data to evaluate the PBL heights?...", "Page 18, line 12: Why not have a figure showing the positive correlation between PBL height and $\delta15N$ ..."
Fig. 9 and Fig. 10 in the revised manuscript

16. Fig 7 can go to the SI.
Confirmed

17. "Fig. 10 is not terribly informative other than to show that the southern boundary should probably also be restricted for the nested area."
The purpose of this analysis is showing how the simulation over the extracted domain improve the performance          (by 1‰ near the southern boundary)

18. I recommend moving Section 3.7 to the first section of the Results.
We decide to start with the interpretation and quantification of the simulation first, then comparing with the measurements.

19. "Strongly recommend putting Figs. 11 and 12 together so that measurements and models are on the same figure. Why not pair the model and obs in time for the figure ..."
Fig. 11 and Fig. 12 shows the characteristic of measurements and simulations, the comparison between them are in the next figure and the associated paragraphs in the manuscript

20. "Fig. 14: Consider normalizing both panels to remove influence of chemistry bias. It seems like there is a signal here that the CMAQ modeling is able to capture, but it is hard to tell."
We do agree the normalization could make the figure looks more beautiful. However, the similar monthly variation and seasonal trend with the gap between the simulated d15N(NOx) and measured d15N(NO3-) is exactly what we are expecting, since there are isotope effects associated with the photochemical transformation of NOx into NOy. After including the 15N into the chemical mechanism in our future work, this gap will be resolved.

---

## Author Response (AR2)

Report 1

In addition to the minor comments below, I have one main reservation remaining. The language is often awkwardly phrased or unnecessarily verbose throughout the text and needs some more polishing. It doesn't interfere with communicating the meaning of the science but streamlining the details and focusing on the main points would crystallize the main messages better. I estimate 25-30% of the text could be cut down or moved to the SI. That said, I understand the authors want to move along to the full-science study with chemistry.

Adjusted accordingly

1. P2L5: "What is meant by "gasoline and diesel in vehicle emissions?" I think some words are missing from this sentence.

Changed to emissions from gasoline- and diesel-powered engine

2. P2L20-24: It's not clear from the abstract how SMOKE only can yield a prediction of concentration of isotope. Consider deleting and focusing instead on…

Rephrased

3. P4L2: Consider revising "N2 = 0.0036" to "(15N2/14N2)air = 0.0036". Is that correct?

Yes

4. P5L3: Recommend rewriting: "simulations by the CMAQ (Community Multiscale Air Quality) modeling system."

Confirmed

5. P5L5: Fang et al., 2021 is not in the citations list.

Deleted, this paper hasn't been published yet.

6. P5L11: Please rewrite to "The EPA trace pollutant emissions mode…"

Confirmed

7. Fig. 2 caption: there is no light purple in t Fig. 2.

Fixed, renamed the color to "light grayish purple"

8. Equation 4: Is 't' the vehicle travel time? Please specify in the text. Does the onroad diesel use the same equation for its isotope contribution or are there different parameters? Does t vary throughout the simulation or is -2.7 assumed constant throughout? I didn't quite understand from the text.

Rephrased

9. P9L29-30: replace 'blur' with 'disperse' and 'blurring' with 'dispersion'.

Confirmed

10. P9L36-P10L6: This paragraph is a bit awkward. I recommend starting out mentioning that chemistry will be explored later. Then define what you mean by "enhanced deposition". Then mention that you assume no difference in the deposition of the isotope and the stable NOx. The

explanation of the impact of deposition on hypothetical powerplant emissions is fine to remove, in my opinion.
Adjusted accordingly

11. Sections 2.2.1-2.2.4: These sections are organized and titled in alignment with CMAQ in puts and processes but often sensitivity runs are mentioned within these sections without being explicitly introduced. I recommend creating a section 2.2.5 that explicitly lists and describes every simulation. Also, the names of these simulations vary.
A summary table is included

Sometimes the first one is called "emission only" and sometimes it's called "no transport".
Fixed, stick to "emission only"

12. P10L32: A clean atmosphere is not necessarily ideal. Was the model provided time for 'spin-up' or initialization so that evaluation against observations would be more appropriate?
The initial concentration is based on the default setting of CMAQ, we only incorporated 15NOx to the input files, using [15NOx] = 0.0036[14NOx]

13. Section 2.2.3: Please include a sentence to explain why chemistry needed to be emulated with enhance deposition. The paragraph transitions from deposition to chemistry abruptly and needs some more detail there.
Confirmed

14. P11L6: "2.2.4"? →2.2.2

15. P11L8: This is not a complete sentence. Typo fixed

16. Fig. 3: Is this tons/day or tons N/day? The text on page 12 uses both units. I suggest sticking with one if possible. tons N/day

17. Fig. 8: I don't think the inset for Indiana is discussed. Why is it so dramatically different from the results in the same location for the larger domain?
Because the interval of the d15N values for zoom-in view is narrower. The purpose for using narrower interval is to show more details in d15N values in Indiana. We are not going to discuss the d15N values over Indiana, just to give the reader more convenience for reading this figure.

18. Fig. 16: I'm glad this comparison is here. Please make sure the x and y axes are equivalent for at least the last figure, but preferably for all of these panels. It will be interesting to see what happens with realistic deposition and chemistry.
Adjusted

19. Fig. S19: Consider promoting to the manuscript. I can't tell what symbols are the NADP values I should be looking at.
Moved the whole paragraph to supplementary

20. PBL height analysis: I'm glad this was added to the study. The discussion is rather vague

though – the authors just ascribe large isotope effects to the stronger effect of atmospheric processes. Can you please expand on this? Are these point source emissions that are being mixed back down to the surface? Or some other explanation?

Detail explanation included

21. P31L18-19: How are the authors concluding that uncertainties in the simulation are less than 5 part per mil? Do they mean variability instead of uncertainty? Please explain how uncertainty is estimated in the text.

Rephrased

Report 2
There are number of errors and a need for some better organization for the manuscript to be publishable, as these do change the scientific meaning of the results in several places. Of key importance is that authors do a better job of distinguishing NOx versus NO versus NO2. This needs to be made very clear upfront in the manuscript and if the authors are going to choose to use NOx as a shorthand when referring to NO or NO2 then that needs to be spelled out specifically. NO is the primary emission from all sources in this work, except potentially diesel emissions. This is never mentioned nor discussed and has an important context here since there are no direct observations of d15NO alone and what the model is actually simulating is the release of NO, not the release of NOx. This should be rectified if the authors want the atmospheric chemistry community to be interested in this work.
Adjusted accordingly

Abstract -
It should be made clear in the abstract the difference between SMOKE and CMAQ (i.e. that the latter includes mixing and transport). This will help setup the reader for why these two models are used and what differences can be expected.
Fixed

Spell out NADP since this is the first use and also make clear that these are rainwater nitrate measurements.
Confirmed

Introduction-
Page 3, Line 3: The very first word of the introduction should not be a symbol – define nitrogen oxides first, then use NOx subsequently.
Confirmed

Line 16: It seems strange to give a single percent (15%) and then give a range of values for soil NOx emissions. It would be better to introduce the range in total NOx emissions from global models and then point out the uncertainty range associated with individual sources such biogenic soil emissions. Also, it should be explained what soil emissions represent (ie. Nitrification and denitrification.
Fixed

Line 17-18: Satellite observations should be mentioned as reference cited in this sentence include these.
Confirmed

Line 24: change "while acting" to "versus acting" or similar, otherwise the sentence is incomplete
Confirmed

Line 40-41: Natural versus anthropogenic sources is not correct here as anthropogenic sources themselves has distinct isotopic range. I suggest you remove "natural and anthropogenic" from the sentence

Confirmed

Page 4, line 11: change "implement" to "implementation"
Confirmed

Line 20: add "species" after d15Noy
Confirmed

Figure 1 caption: CFPP needs to be defined in the figure caption.
It has been defined, coal-fired power plant (CFPP)

Page 5, line 4-7: might make sense to have this sentence be past tense since that paper is already published.
Confirmed

Throughout section 2 and its subsections there needs to be consistent use of n values, consistent use of ranges versus average values, consistent discussion of methodologies and there needs to appear a discussion of NOx versus NO2 versus NO. Many of the measurements are not able to be compared to one another as some measure NOx and some measure NO2.
Fixed

Page 7, line 1-4: Miller et al and Yu & Elliott both use active sampling techniques for collecting soil emissions. Not clear what "using a similar methodology at the end of Line 1 means, sinc the previous study mentioned used a passive sampling technique. Further, the passive measurements should report an n value and not a range since I believe these two numbers represent two discrete measurements that represent month long averages.
Fixed

Section 2.1.2 – it needs to be justified why Miller et al (2017) is not included. This was brought up in the previous review both reviewers. This study is an actual on-road study that captures the mix of tailpipe emissions. The Walter's study is more representative of tailpipe type measurements, not the signal that is found on-road when vehicles are moving under real traffic conditions with a mix of vehicle types. The Miller study seems more representative in this case. If it is not, it should be reasoned why other values are being used.
Fixed

Line 22: remove "s" in this sentence
Confirmed

Page 8, section 2.1.3: this section does not make mention of the methods used, nor does it distinguish what was measured in terms of NOx vs NO2.
The sum of NO and NO2

Line 19-20: how can a range of 9 to 26 lead to an average of 4.51? Please double check.
It has been written as "standard deviation"

Page 9, line 36: change the terminology of "d15NOx effect" as this is not an accurate statement. It's really the effect on d15NOx
Confirmed

Line 37: unbold "not"
Confirmed

Page 10, line 6: change "the" to "a"
Confirmed

Page 11, line 3: change "to" to "in"
Confirmed

Page 14, line 2-3: please also define the anthropogenic fraction since that is used several times in the text below
Confirmed

Line 21: change "the clear" to "a clear"
Confirmed

Line 20-24: justify to the reader why your are selecting April to June to show here (I assume b/c this is the season when soil NOx emissions should be maximum?)
Clarified, highest variation during April to June

Page 15, Figure 5 caption: explain that the white box is defined in the Figure 6 caption.
Confirmed

Page 18, line 8: change here since April to October is not a "month"
Confirmed

Page 20, line 20: change to "a seasonal"
Confirmed

Line 22: change to "NOx was emitted"
Confirmed

Line 31: this line does not make sense, please rephrase.

Page 20, Figure 8: draw a line to make clear that the insert is Indiana (ie draw a line to the state to connect it) or label the smaller box just above as Indiana to make clear to the reader.
Confirmed

Page 21, line 18: This should start a new section as the emphasis here is no longer on seasonality and rather on the impact of different meteorological fields.
Confirmed

I would argue that Figure 9 and 10 could be combined with a different coloring or dashed lines to represent the different years of met fields.
Moved Fig. 10 to supplementary

Page 22, line 4: add "the" before simulation
Confirmed

Figure 11 is unnecessary as the text described this very well and the changes are smaller than even the analytical reproducibility of measurements so there is no way to even test where this distribution is realistic (and the boundary lines are clearly not realistic/an artifact of the model domain).
Moved to supplementary

Page 24, Section 3.5: the phrasing here would be better as NOx loss rather than NOx deposition. The increase in NOx deposition is purely artificial – i.e. this is not, for instance, testing a range in the literature to look at sensitivity. It is a change in the amount of NOx loss to simulate what happens to the isotopic composition given NOx loss from the atmosphere.
Confirmed

Page 25, line 7: remove "of atmospheric"
Confirmed

Line 9: there is no Figure 12a. Figure 12 is not really utilized in this section and it is never explained why a particular time point in the model is chosen for this comparison. Either robustly explain the purpose of the figure and the conclusion being drawn from it or remove it.
Fixed

Line 10: change "as a similar fashion" to "in a similar fashion"
Confirmed

Line 17: change to "lead to lower"
Confirmed

Line 19-20: change to "atmospheric NOx on a seasonal basis"
Confirmed

Line 20: the range here says a maximum of 5.34 per mil but the figure range only goes to 4 per mil?
Fixed

Line 25-27: the sentence beginning with "The enhanced deposition simulation somehow presents the isotope effects…." This sentence is not accurate. This cannot represent the fractionations associated with chemistry. What it does show should be the focus here. And it should be explained here that this is used as a proxy for NOx loss from the atmosphere as would be the case in conversion to NOy species. But this is not at all representative of how and why chemical processes would lead to fractionation of NOy species.

Fixed

Page 26, line 4: change "comparing" to "compared"
Confirmed

Page 27-28, Section 3.6: the two paragraphs of this section are repetitive. This section should be re-organized. I would suggest making the section on comparison to observations its own as Section 4, etc. And the beginning of the section should explain what observations will be used for comparison purposes with the model. It should also be explained what the observations represent and how NOx was computed from d15NO2 measurements. It should also be explained by isotopic measurements of nitrate in rainwater from NADP sites is used as a comparison here as well. Then move onto the actual comparisons.
Fixed

Page 28, line 20: change to "based on 2002"
Confirmed

Page 29, line 1: "collect" should be "collected"
Confirmed

Line 3: change to "show similar monthly variations"
Confirmed

Line 10: chage to "NO3- are about"
Confirmed

Page 30: The paragraph below Figure 16 belongs before the previous paragraph and these two paragraph are repetitive and should be edited further.
Fix

Figure 16: It's not clear what is being gained from the Figure 16 comparison. The text focuses on discussion of how the seasonal variability is captured. Clearly the ranges are not always captured and while in some cases the variability is decently captured (i.e. R2>0.4) we are not really learning anything by comparing the data in this way. And it is not discussed as to why some comparisons are good and others are not at all. How can we expect the model to ever get d15NO3- values exactly right when chemistry is not even included in the model? It would make a lot more sense to compare seasonal averages or time series from the model versus time series of the observations and focus on the relative changes from season to season rather than leading the reader to expect that there should be a 1:1 relationship for the model-observation comparison. Further the last line of this section (page 31, line 10) is not appropriate speculation. There's nothing to justify this statement and again the model is not simulating nitrate fields.
Fix

Page 31, Conclusion:

The discussion of "uncertainty" needs to be clarified. This is not addressed earlier in the text, so

it is not clear how an uncertainty of less than 5 per mil (line 18) was determined. The word uncertainty does not make sense in the context here. And discussion of uncertainty is also unwarranted since the model does not consider/test the full ranges of the observed NOx source signatures for different emission sources. In fact the word uncertainty is used again to describe the uncertainty in NOx emission, which is documented in the literature. Really what is being discussed is the narrow range predicted by the model – or so it appears to me.

Rephrased, the uncertainties here refers to the d15N values in emission input dataset, rather than the uncertainties of the prediction of the model.

Line 28: this sentence makes no sense and it is never discussed in the text that there is such a large uncertainty in NOx power plant emissions.
Reference added

Line 35 and 37: change the phrase "are possible to be"
Confirmed

Line 37: change "estimated" to "predicted" and add "to be" following NOx
Confirmed

Line 44: what constitutes a "better" simulation? Ie. Based on what is it deemed better?
Rephrased